# Intravital deep-tumor single-beam 3-photon, 4-photon, and harmonic microscopy

**Gert-Jan Bakker[1]\*, Sarah Weischer[1], Júlia Ferrer Ortas[2], Judith Heidelin[3], Volker Andresen[3], Marcus Beutler[4], Emmanuel Beaurepaire[2], Peter Friedl[1,5,6]\***

[1]Department of Cell Biology, Radboud Institute for Molecular Life Sciences, Radboud University Medical Centre, Nijmegen, Netherlands; [2]Laboratory for Optics & Biosciences École Polytechnique, CNRS, INSERM, Paris, France; [3]LaVision BioTec GmbH, a Miltenyi Biotec company, Bielefeld, Germany; [4]APE Angewandte Physik & Elektronik GmbH, Berlin, Germany; [5]Cancer Genomics Centre, Utrecht, Netherlands; [6]David H. Koch Center for Applied Genitourinary Cancers, The University of Texas MD Anderson Cancer Center, Houston, United States

**\*For correspondence:**
gert-jan.bakker@radboudumc.nl (GJB);
peter.friedl@radboudumc.nl (PF)

**Abstract** Three-photon excitation has recently been demonstrated as an effective method to perform intravital microscopy in deep, previously inaccessible regions of the mouse brain. The applicability of 3-photon excitation for deep imaging of other, more heterogeneous tissue types has been much less explored. In this work, we analyze the benefit of high-pulse-energy 1 MHz pulse-repetition-rate infrared excitation near 1300 and 1700 nm for in-depth imaging of tumorous and bone tissue. We show that this excitation regime provides a more than 2-fold increased imaging depth in tumor and bone tissue compared to the illumination conditions commonly used in 2-photon excitation, due to improved excitation confinement and reduced scattering. We also show that simultaneous 3- and 4-photon processes can be effectively induced with a single laser line, enabling the combined detection of blue to far-red fluorescence together with second and third harmonic generation without chromatic aberration, at excitation intensities compatible with live tissue imaging. Finally, we analyze photoperturbation thresholds in this excitation regime and derive setpoints for safe cell imaging. Together, these results indicate that infrared high-pulse-energy low-repetition-rate excitation opens novel perspectives for intravital deep-tissue microscopy of multiple parameters in strongly scattering tissues and organs.

## Editor's evaluation

Nonlinear microscopy is in the unique position that high-resolution images of cells and other tissue components can be obtained in live tissue. However, scattering and absorption limit the penetration depth and thus the level of 3D information that can be achieved. In this manuscript, the authors develop a new approach that can accomplish deeper imaging in complex specimens, which has potential to provide new insight into biological mechanisms within living tissues and whole organisms.

## Introduction

Intravital microscopy enables studies of the physiology and malfunction of live cells in multicellular organisms (*Ritsma et al., 2012*). Several excitation and detection modalities have been applied for multiparameter three-dimensional (3D) imaging in live tissues, yielding different compromises

between imaging depth and resolution. Linear imaging modalities such as confocal microscopy enable a penetration depth less than 100 µm (*Helmchen and Denk, 2005*), which restricts their application to low-scattering, thin or surface-specific tissue models, such as the mouse inner ear and chick embryo (*Deryugina, 2016*; *Tomo et al., 2007*; *Figure 1a*, *Supplementary file 1a*). Taking advantage of reduced scattering and absorption of the excitation light and selective excitation of the focal plane (*Helmchen and Denk, 2005*; *Diaspro et al., 2005*), 2-photon (2P) microscopy based on high-frequency near-infrared excitation pulses (80 MHz, ~100–200 fs, 800–1300 nm, further referred to as lowIR) provides 100–400 µm imaging depth in highly scattering tissues, for example, liver (*Ritsma et al., 2012*) and skin (*Andresen et al., 2009*), and up to 500–800 µm in less scattering tissues, such as mouse brain cortex (*Helmchen and Denk, 2005*).

With increasing imaging depth in tissue, 2P imaging requires to increase the excitation power in order to compensate for tissue-induced light scattering, absorption and aberrations, to maintain a constant intensity at the plane of focus. However, the required exponential increase in excitation power may provoke light-induced tissue damage (*Débarre et al., 2014*). Heating caused by linear absorption can be minimized by redistributing the excitation power into a reduced number of pulses per second and a higher energy per pulse (further referred to as highIR), which enhances the excitation efficiency per pulse. For example, reducing the pulse repetition rate from 80 to 1 MHz while increasing the pulse energy is an effective means to reach larger depths (*Beaurepaire et al., 2001*; *Theer et al., 2003*), albeit at the cost of reduced imaging speed, because 80 times less excitation events per second are introduced into the sample. Ultimately, 2P imaging depth in densely labeled fluorescent samples is limited by out-of-focus background produced in superficial regions, which at large depths becomes dominant over the signal produced in the focus (*Helmchen and Denk, 2005*; *Theer and Denk, 2006*).

To reduce these confounding effects, shifting the excitation wavelength to the infrared optical windows at ~1300 or 1650–1700 nm was shown to reduce light scattering and, by 3-photon (3P) excitation, improve confinement of the scan focus and thereby reduce out-of-focus background signal (*Xu et al., 1996*; *Hell et al., 1996*). This approach is compatible with commonly used fluorophores (*Deng et al., 2019*; *Hontani et al., 2021*), and was demonstrated to increase penetration in densely labeled brain tissue by 50–100%, reaching imaging depths beyond 1 mm in the brain of live mice (*Horton et al., 2013*; *Ouzounov et al., 2017*; *Yildirim et al., 2019*; *Weisenburger et al., 2019*).

In fundamental and preclinical cancer research, intravital microscopy is an important strategy to record dynamic processes of the tumor microenvironment, including tumor cell invasion, metastasis, and interactions with the tumor stroma. Intravital microscopy of the tumor microenvironment faces several tissue-intrinsic challenges. Tumors are dense, opaque, three-dimensional objects with a size of millimeters to centimeters, which causes strong light scattering (*Andresen et al., 2009*). In addition, analysis of the tumor microenvironment requires to detect multiple cell types, their morphology and activation states simultaneously, and tissue structures over time through a body window (*Weigelin et al., 2021*). HighIR combined with 3P microscopy enables the label-free detection of metabolites, tissue morphology, and leukocyte migration in mammary tumor microenvironment in vivo (*You et al., 2018b*). However, the relevance of 3P intravital microscopy for multiparametric detection of tumors expressing fluorescent reporters, as used in preclinical cancer research, and detection of tumor cells deeply inside the tumor has not been explored. We here apply 1300 and 1650–1700 nm highIR excitation to achieve 3- and 4P and higher harmonics microscopy of live fluorescent tumors in mice and other strongly scattering tissues, such as bone. We report improved multiparameter detection of cells and complex tissues, a 2-fold gain of penetration depth at non-toxic laser intensities, and established criteria to minimize photodamage that enable long-term time-lapse imaging of live-cell biology.

## Results and discussion

### 3P excitation setup characterization

In contrast to the well-established 2P microscopy lowIR excitation regime (100–200 fs pulses with sub-nJ energy density at the focus), efficient 3P microscopy requires shorter (sub-100 fs) pulses at the focus with ~5–10 times higher energy (*König et al., 1999*) (highIR excitation) to generate comparable excitation of fluorophores (*Xu et al., 1996*; *Wang and Xu, 2020*; *Supplementary file 1a*, *Figure 1a*). To keep the average power and associated sample heating at an acceptable level, 3P microscopy

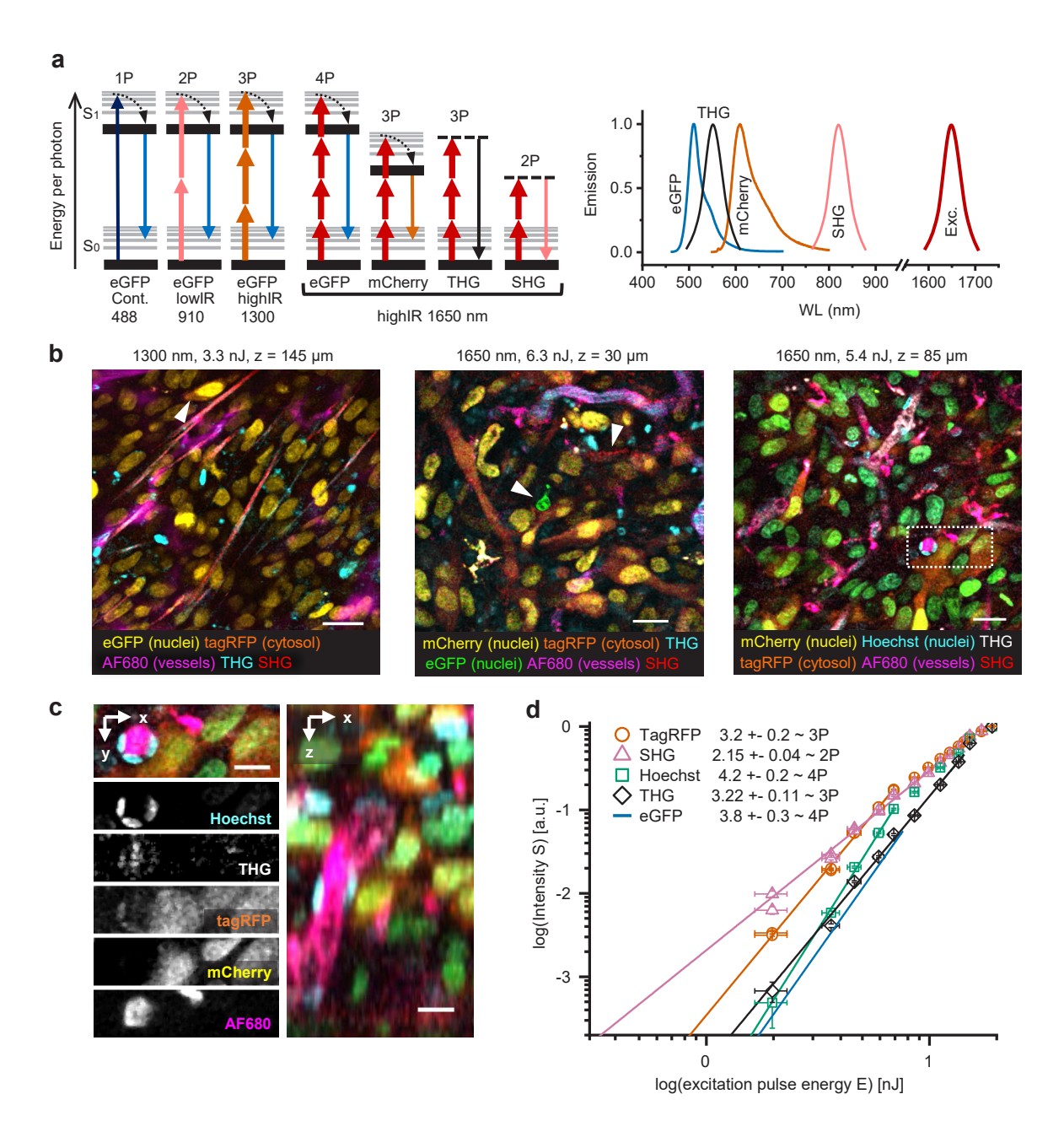

**Figure 1.** Multimodal microscopy of fluorescent skin tumor xenografts in vivo showing up to four fluorescent and two label-free tissue morphology channels, simultaneously excited with a single highIR wavelength. (**a**) Jablonski diagrams (left) and scaled excitation/emission spectra (right) of 1-, 2-, 3-, and 4P-excited fluorescence and higher harmonic generation. Diagram 1–3 (left to right): 1- up to 3P excitation of eGFP with 488 nm continuous, 910 nm lowIR, and 1300 nm highIR, respectively. Diagram 4–7: excitation with 1650 nm results in signals of a distinct emission wavelength, including 4P-excited eGFP. S0 and S1, ground level and first-excited electron state, respectively, with vibrational levels (thin lines). Dashed lines, virtual states. Thin–thick upward arrows: low–high photon density required for excitation. Downward arrows: emission. (**b**) Five-channel 1300 nm excited (left) and six-channel 1650 nm excited (middle and right) images were taken in the center of fluorescent tumors through a dermis imaging window. Images were selected from median-filtered (one-pixel) z-stacks. The excitation wavelength, calculated pulse energy at the sample surface, and imaging depth z are indicated (top). Cell nuclei containing a mixture of mCherry and Hoechst appear as green (right). Details from images left to right: day-10, day-13, and day-14 tumor; 9.2, 4.5, and 4.4 s image acquisition time; 24, 12, and 12 μs pixel integration time; 0.36, 0.46, and 0.46 μm pixel size. (**c**) Zoomed xy-plane with individual channels (left) and orthogonal (xz-) projection (right) from (**b**), dotted rectangle. (**d**) Relation between measured emissions and excitation energy reveals the order of the excitation processes. Normalized emission intensity ($S$) as a function of pulse excitation energy ($E$), for TagRFP, SHG, Hoechst, third

*Figure 1 continued on next page*

*Figure 1 continued*

harmonic generation (THG), and eGFP recorded with an excitation wavelength of 1650 nm. Data was fitted with $S(E) = A \cdot E^n$, with $A$ the proportional factor and $n$ the order of the excitation process (indicated as numbers in the figure). For curve fitting, excitation intensities at the sample surface below the threshold of physical damage (14 nJ) for SHG and THG and up to their saturation limit (7.6 nJ, eGFP; 6.9 nJ, TagRFP and Hoechst) for fluorophores were used. Images were acquired at the same position as panel (a, right), except for the fit line of eGFP, which was retrieved from a different dataset (*Figure 1—figure supplement 4*). Bars, 25 µm (**b**); 12.5 µm (**c**). Source data files: *Figure 1—source data 1* and Figure 1-source data 2.

The online version of this article includes the following video, source data, and figure supplement(s) for figure 1:

**Source data 1.** Numerical data, calculations and templates used to generate graph panels.

**Figure supplement 1.** Microscope setup and excitation parameters.

**Figure supplement 2.** Attenuation of excitation energy at the sample surface by water immersion.

**Figure supplement 3.** Third harmonic generation (THG) and 3-photon-excited fluorescence (3PE) microscopy of the mouse neocortex and external capsule (EC), excited with highIR and lowIR excitation, ex vivo.

**Figure supplement 4.** Two-, 3-, and 4-photon processes excited by 1650 nm highIR excitation.

**Figure 1—video 1.** Single-beam 3-photon, 4-photon, and harmonic microscopy of fluorescent tumor xenograft in vivo, excited with 1650 nm.
https://elifesciences.org/articles/63776/figures#fig1video1

applies lower pulse repetition rates (1–2 MHz) compared to pulse repetition rates commonly used for 2P microscopy (80 MHz) (*Beaurepaire et al., 2001*; *Theer et al., 2003*). The source used in this study consisted of a fiber laser pumping an optical parametric amplifier (OPA) providing 1 MHz, sub-100 fs pulses at 1300 or 1650 nm wavelengths. Excitation was focused with a high-NA objective, resulting in a resolution better than 0.8 µm in the lateral and 3 µm in the axial dimension (*Figure 1—figure supplement 1*). Physiological saline was used as immersion medium for high compatibility with the tissue and observation window. Since part of the excitation pulse energy ($E_{obj}$) was absorbed by the immersion medium, we calculated the pulse energy at the sample surface ($E$) as a reference for each experiment (*Figure 1—figure supplement 2*). We compared highIR excitation at 1300 and 1650 nm with lowIR excitation in the range 1100–1300 nm, the current standard for in-depth multiphoton imaging (*Andresen et al., 2009*). Using the mouse brain as a reference (*Horton et al., 2013*; *Weisenburger et al., 2019*), we confirmed that 1650 nm highIR excitation enables to resolve structural details with third harmonic generation (THG) contrast at depths encompassing the entire cortex (>1 mm), which outperforms the imaging depth obtained with 1270 nm lowIR excitation by almost a factor of two (*Figure 1—figure supplement 3b,c*). Since both high- and lowIR excitation measurements were using a 3P process to generate contrast, the superior performance with highIR excitation can be attributed to the enhanced excitation efficiency rather than to reduced out-of-focus background. We calculated the effective attenuation length ($l_e$), which is the mean distance travelled by light before being scattered or absorbed by the tissue (*Wang et al., 2018b*). In the neocortex, $l_e$ was 336 µm for 1650 nm excitation and 280 µm for 1280 nm excitation (*Figure 1—figure supplement 3d*), in agreement with previous work (*Horton et al., 2013*; *Wang et al., 2018b*).

## Simultaneous multicolor 3P, 4P, and harmonic generation microscopy with a single excitation beam

When applied to multicolor-labeled HT-1080 sarcoma tumors in the deep dermis, excitation with 1300 or 1650 nm highIR pulses produced simultaneous distinct signals from fluorescent proteins (eGFP, TagRFP, mCherry), a far-red vascular tracer (AlexaFluor680-conjugated 70-kD-dextran) and blue fluorescence (Hoechst 33342), together with second harmonic generation (SHG) and THG signals (*Figure 1b–c*, *Figure 1—video 1*). To understand the nonlinear processes underlying this unusually broad spectrum of fluorophores excited with a single laser line, we investigated the order of the excitation process for each signal (*Figure 1a*). We recorded the emission intensity in response to increasing excitation energy $E$ and estimated the order of dependence between emission intensity and $E$ by fitting the data to a power law (*Figure 1d*, *Figure 1—figure supplement 4* and *Supplementary file 1b*). As internal controls, SHG and THG signals were used as reference for second- and third-order processes (*Cheng et al., 2014*). When excited at 1650 nm, the red fluorophores TagRFP, mCherry, and AF680 followed a cubic dependence on excitation power, while green and blue fluorophores eGFP and Hoechst followed a quartic dependence, consistent with respective third- and fourth-order processes (*Cheng et al., 2014*). This shows that 3- and 4-photon-excited fluorescence (3PEF, 4PEF)

along with multi-harmonic SHG and THG signals were obtained simultaneously using 1650 nm excitation, resulting in six-channel images in a single scan (*Figure 1b*). Although independent control of each respective signal strength is not possible, a key advantage of this one-shot excitation scheme is that it is free of chromatic aberration, a common problem in multi-beam imaging (*Theer et al., 2014*).

## Characterization of phototoxicity and bleaching

The high pulse energy required for efficient highIR excitation is a double-edged sword. Although it enables multi-modal excitation as seen above, pulsed light can induce nonlinear and linear perturbations. First, the energy density within the beam focus, which causes nonlinear interactions between a strong electric field and matter, may induce physical damage by ionization (*Konig et al., 1997*), toxic reactive oxygen species, and photobleaching (*Débarre et al., 2014*; *Tirlapur et al., 2001*). This focused energy may further cause local transient heating of the focal volume during the excitation pulse, caused by water absorption (*Qiu et al., 2017*). This type of phototoxicity is dependent on the pulse energy (nJ) at the focal plane. Second, phototoxicity may arise from linear absorption of light above and below the focal plane, which may cause accumulation of heat in a larger tissue volume (*Débarre et al., 2014*; *Podgorski and Ranganathan, 2016*; *Wang et al., 2018a*; *Rowlands et al., 2017*). Sample heating is proportional to the average power of the incident excitation beam. Previous studies have investigated phototoxicity mechanisms with 1300 nm highIR excitation in the brain, and have established safety thresholds for 3P imaging of neuronal activity in this wavelength range (*Ouzounov et al., 2017*; *Yildirim et al., 2019*; *Wang et al., 2020*). In this work, we investigated the mechanisms and onset of photoperturbation with 1700 nm highIR excitation in tumor models in vitro (tumoroids), and with 1300 and 1700 nm highIR excitation in mouse dermis tumors in vivo and achieved safe live-cell imaging conditions.

As a readout for cell stress, we continuously recorded the intracellular $Ca^{2+}$ responses of live tumor cells expressing a $Ca^{2+}$ sensor GCaMP6 (*Chen et al., 2013*) in combination with the cell-death indicator Sytox Green (*Figure 2a*). GCaMP6 fluorescence was monitored using lowIR 2P excitation (920 nm, 80 MHz) in parallel to 1700 nm highIR exposure (see *Materials and methods*). Intracellular $Ca^{2+}$ responses represent a sensitive readout for non-lethal cell stress, caused by opening of calcium channels as well as by transient membrane damage caused by reactive oxygen species and other cytotoxic events (*Hopt and Neher, 2001*; *Iwanaga et al., 2006*). In a collagen-based 3D phantom containing a reconstituted cell-rich tumoroid, spontaneous reversible $Ca^{2+}$ fluctuations and few Sytox-positive damaged cells were detected before highIR exposure (*Figure 2b*, frame #70). As a positive control, intermittent highIR scanning at toxic intensity (17 nJ) caused rapid-onset $Ca^{2+}$ elevation in cell subsets followed by gradual increase of the Sytox signal (*Figure 2b*, frames #102–201). High exposure was further confirmed by significant photobleaching of mCherry emission from cell nuclei (*Figure 2b*, arrowheads). After termination of toxic highIR exposure, the $Ca^{2+}$ elevation persisted, and an increasing number of cells became Sytox-positive (*Figure 2b*, frame #425). To study the toxicity of highIR exposure in detail, Sytox-positive and -negative cells at the endpoint (frame #425) were grouped and the kinetics of $Ca^{2+}$ and Sytox responses of both subsets were quantified (*Figure 2c* and *Figure 2—figure supplement 1a*). The Sytox-negative, surviving cells showed a transient $Ca^{2+}$ elevation which declined to half of the maximum amplitude after cessation of highIR exposure, whereas Sytox-positive cells showed an accelerated and transient $Ca^{2+}$ elevation upon onset of highIR exposure in combination with delayed nuclear Sytox increment (*Figure 2c*). In Sytox-positive cells, the GCaMP6 signal level eventually declined to its initial level before highIR exposure, likely as a consequence of persisting membrane damage with leakage of GCaMP6 in parallel to Sytox entry. At the single-cell level, $Ca^{2+}$ elevation was nearly instantaneous with the onset of laser exposure in Sytox-positive cells, followed by Sytox elevation thereafter with a large spread of the onset time (*Figure 2—figure supplement 1b*). Thus, the $Ca^{2+}$ signal reliably indicated phototoxic stress, prior to detection of cell death by Sytox and even in cells which remained viable at the experimental endpoint.

Using both GCaMP6 and Sytox recording, we then aimed to identify the highIR conditions tolerated by live tumor cells upon consecutive scanning, using a highIR energy escalation approach followed by quantification of the highIR-induced $Ca^{2+}$ responding (GCaMP+) cell fraction for each condition (See *Materials and methods* section for further details). For moderate highIR pulse energies, comprising pooled datasets from 5.0 and 8.4 nJ, the median percentage of cells with elevated $Ca^{2+}$ was 0.7%, whereas at 12 nJ this fraction was 10-fold increased (*Figure 2d*). Thus, a threshold for excitation

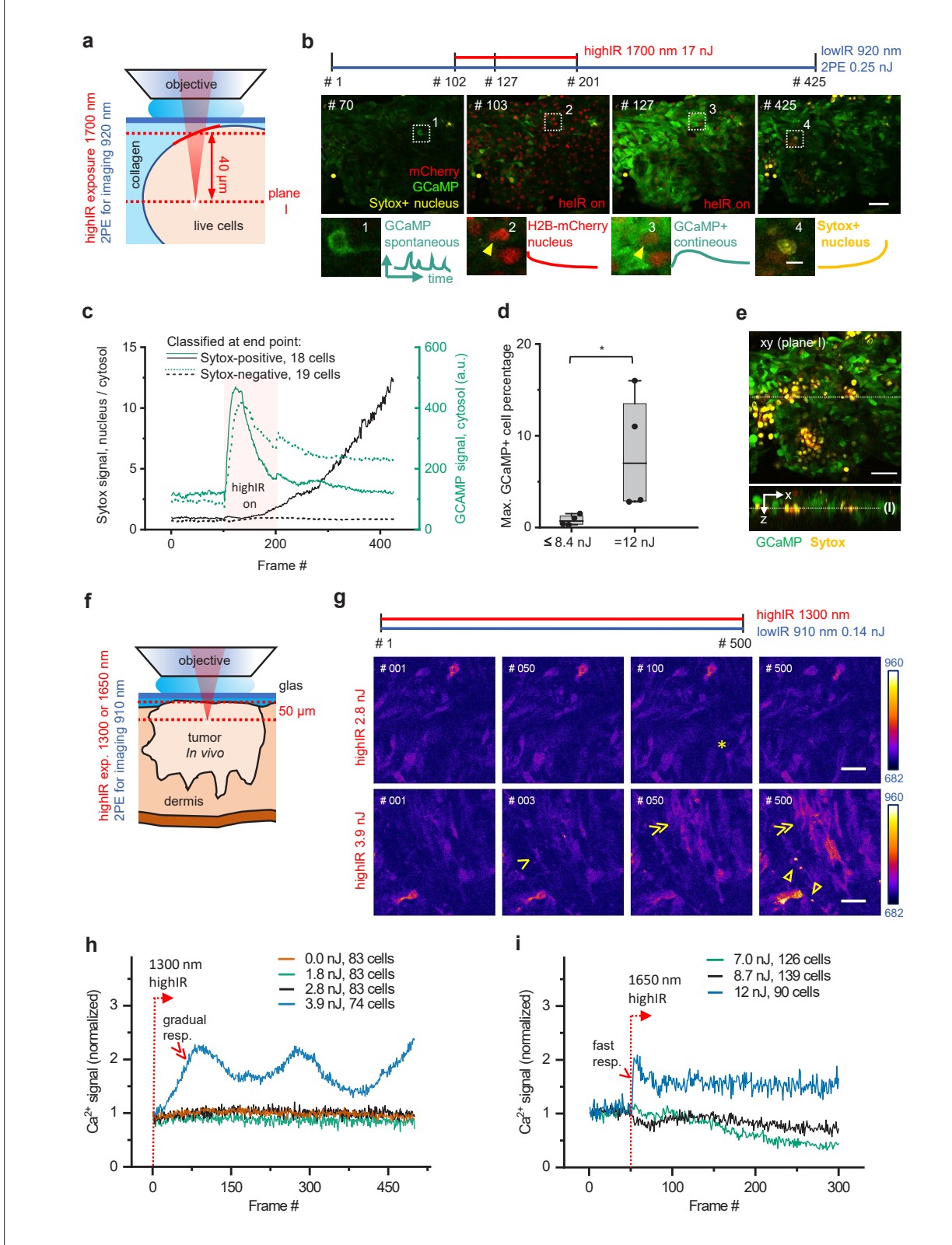

**Figure 2.** Phototoxicity and interference with biological function induced by continuous scanning with highIR excitation. (**a**) Experimental approach for cell viability detection in vitro. B16F10/GCaMP6/H2B.mCherry (**Chen et al., 2013**) melanoma tumoroid embedded in 3D collagen was monitored through a coverglass. Sytox Green was supplemented to the medium. Both GCaMP and Sytox signals were continuously monitored (3.3 s/frame, 6 μs pixel integration time) at ~40 μm penetration depth using low pulse-energy excitation (920 nm). During part of the measurement, the sample

*Figure 2 continued on next page*

*Figure 2 continued*

was simultaneously exposed to 1700 nm highIR excitation, approximately at the same focal plane (**l**). ( **b**) Representative frames from continuous scanning with the indicated excitation pulse energy and detection of emitted signals. Details from dotted squares: (1) spontaneous reversible $Ca^{2+}$ activity (green); (2) highIR-excited fading mCherry signal from the nuclei (red); (3) strong highIR-induced $Ca^{2+}$ response in the majority of the cells and bleached, decreasing mCherry signal; and (4) Sytox-positive nucleus (yellow). (**c**) Intensity curves of Sytox (black, left y-axis) and GCaMP (turquoise, right y-axis) signals in cells developing Sytox-positive nuclei (solid lines) or not (dashed lines), classified at the endpoint by eye (frame #425). Data show the mean intensity of single-cell intensities. For technical details, see *Figure 2—figure supplement 1a and b*. (**d**) Predictive power of GCaMP positivity for photodamage. The highIR-induced percentage of GCaMP-positive cells was quantified over time for moderate- (≤8.4 nJ; ~1220 cells from n = 4 measurements) and high-exposure (12 nJ; ~720 cells from n = 4 measurements) conditions. *, p = 0.03, Mann-Whitney test. Data indicate the median (line), 25/75 percentile (box) and 5/95 percentile (whiskers). (**e**) Spatial restriction of GCaMP and Sytox positivity to excitation plane (**l**). A z-stack of 80 µm depth (5 µm step size) was taken 15 min after the endpoint of highIR exposure shown in (**b**). Above: xy-image at the highIR exposure plane l. Below: xz-sideview positioned at the line in the xy-image. Line in sideview: excitation plane l. (**f**) Detection of phototoxicity of highIR in vivo. Intradermal tumor imaged after 6 (1650 nm) or 11 days (1300 nm) of growth. GCaMP signal was continuously monitored (~1.5 s/frame, 6 µs pixel integration time) in an ~50 µm deep focal plane using lowIR excitation (910 nm). The sample was simultaneously exposed to highIR excitation at the same focal plane. (**g**) $Ca^{2+}$ signaling in individual cells. Representative frames from the 1300 nm highIR exposure time-series at pulse energies at the sample surface below (2.8 nJ), or above (3.9 nJ) the toxicity threshold. Asterisk, spontaneous, reversible $Ca^{2+}$ signal in single cell, as seen in ~10 of 83 cells in the field of view. Arrowhead, persistent $Ca^{2+}$ signal starting at frame #2, as seen in 8 of 74 cells. Double arrowheads, multiple cells developing increasing $Ca^{2+}$ signal, present in 45 of 74 cells. Closed arrowheads, burning marks. The frame number is indicated. (**h, i**) $Ca^{2+}$ signal as a function of time for increasing highIR pulse energies at the sample surface, recorded with an excitation wavelength of 1300 nm (**h**) or 1650 nm (**i**). Single and double arrowheads: steep and gradual $Ca^{2+}$ rise, respectively, related to cell populations as described in (**g**). Emission signal was retrieved after averaging over the image area, background subtraction and normalization to the first highIR-excited frame (#1, 1300 nm and #50, 1650 nm, dotted lines). The number of cells per field is indicated. Bars: 50 µm, 10 µm (details). Source data files: *Figure 2—source data 1* and Figure 2-source data 2.

The online version of this article includes the following video, source data, source code, and figure supplement(s) for figure 2:

**Source code 1.** Conversion of fiji manual tracking data into intensity-time traces.

**Source code 2.** Raw microscopy image files, intermediate and analysis results.

**Source code 3.** Retrieve cell fraction responding upon highIR exposure as a function of time.

**Source code 4.** Retrieve cell fraction responding upon highIR exposure as a function of imaging depth.

**Source data 1.** Numerical data, calculations and templates used to generate graph panels.

**Figure supplement 1.** Methodological strategy of cell viability measurements in vitro.

**Figure supplement 2.** Scanning-induced fluorophore bleaching and stasis of vascular perfusion.

**Figure 2—video 1.** Phototoxicity and interference with biological function induced by 1300 nm highIR excitation, in vivo.
https://elifesciences.org/articles/63776/figures#fig2video1

**Figure 2—video 2.** Phototoxicity and interference with biological function induced by 1650 nm highIR excitation, in vivo.
https://elifesciences.org/articles/63776/figures#fig2video2

energy in the focus of up to 8.4 nJ at the sample surface allows for safe time-lapse imaging inside the tissue, when focusing at least 40 µm deep inside the sample.

To account for linear phototoxicity outside of the highIR focal plane, depth stacks were obtained before and after time-lapse measurements at low-to-high highIR exposure dose (5.0–17 nJ). Both $Ca^{2+}$ and Sytox signals were spatially restricted within the highIR excitation plane, without effects in regions above and below (*Figure 2e* and *Figure 2—figure supplement 1c-f*). This result shows that continuous scanning with 1700 nm highIR excitation power of 17 mW (17 nJ × 1 MHz) at the sample surface can be used without causing linear phototoxicity.

To verify the setpoint for phototoxicity for intravital microscopy, we exposed mouse dermis tumors to 1300 and 1650 nm highIR excitation and monitored their cellular $Ca^{2+}$ responses (*Figure 2f*). During continuous highIR exposure for energies at the sample surface below or equal to 2.8 nJ (1300 nm) or 8.7 nJ (1650 nm), the $Ca^{2+}$ signal retained background activity, with occasional spontaneous $Ca^{2+}$ fluctuations (*Figure 2g*, asterisk; *Figure 2—videos 1; 2*). Higher excitation energies (3.9 nJ for 1300 nm, 12 nJ for 1650 nm) induced $Ca^{2+}$ elevation in cell subsets (*Figure 2g* and *Figure 2—videos 1; 2*, arrowheads). These $Ca^{2+}$ responses differed from the background fluctuations by their steep or gradual increase of signal (*Figure 2h and i*, arrowheads). Moreover, at prolonged exposure above the observed thresholds, $Ca^{2+}$ signal induction preceded the onset of burning marks (*Figure 2g* and *Figure 2—video 1*, *Figure 2—video 2*, closed arrowheads) or intravascular blood stasis (*Figure 2—figure supplement 2a*). Furthermore, to avoid thermal damage induced by heating, we applied average power levels at the sample surface of at most 38 mW (38 nJ

times 1 MHz) under live-cell and in vivo conditions, which in the brain suffices to limit tissue heating below ~1.8°C (*Podgorski and Ranganathan, 2016*; *Rowlands et al., 2017*). Thus, we established a limit for power densities to be used for multimodal excitation in tumors to remain below functional phototoxicity levels and showed that higher doses induce different grades of damage (*Iwanaga et al., 2006*; *Koester et al., 1999*).

To quantify photobleaching, repeated 3D scanning was performed over 60–75 min (one 3D scan/min), using highIR exposure within the toxic (3.5 nJ, 1300 nm; 12 nJ, 1650 nm) or nontoxic range. Within the toxic range, eGFP intensity declined by 25% after 25 scans (1300 nm) and mCherry intensity declined by 25% after two scans (1650 nm) (*Figure 2—figure supplement 2b*). Within the non-toxic range, mCherry intensity decreased by 10–20% after 50 scans, while 3P-excited eGFP signal remained stable. This level of photobleaching may limit applicability for imaging with sub-second frame rates, as required for $Ca^{2+}$ monitoring in the brain (*Weisenburger et al., 2019*), but remains acceptable for monitoring with lower frame rates, in the minute-range up to days, and for large-volume scanning, which are relevant imaging regimes for monitoring tumor cell kinetics and therapy response in tumor models (*Haeger et al., 2020*; *Khalil et al., 2020*).

## Deep-tumor multiparameter microscopy with highIR excitation

We next investigated whether highIR excitation at 1300 and 1650 nm provides an advantage for imaging deep-tumors regions, with respect to conventional lowIR excitation at 1180 nm using a titanium sapphire/optical parametric oscillator (Ti:Sa/OPO) combination (*Figure 3*). To achieve arbitrarily constant emission with increasing imaging depth, we manually increased the excitation energy *E* using eGFP (for 1300 nm) and TagRFP (1180 and 1650 nm) as a reference, while remaining below the phototoxicity thresholds as defined above (*Figure 3a*, gray profiles). 3P and 4P highIR excitation resolved subcellular features of eGFP and TagRFP labeled cells at depths beyond 400 µm, as compared to 225 µm for OPO-generated 2P lowIR excitation (*Figure 3a*). Multiparameter recordings were reliably obtained at a depth of 350 µm inside the tumor using 1300 and 1650 nm highIR excitation, but not with the 1180 nm lowIR source (*Figure 3b*). To investigate the imaging depth limit for the different modalities, we first quantified the signal-to-noise ratio (SNR) of fluorescent features as a function of imaging depth. While the SNR remained above the detection limit (SNR ≥3) until a depth of only ~255 µm using the 2P lowIR excitation regime, the 3P highIR excitation regime enabled us to reach depths of ~395 µm (1650 nm) or ~415 µm (1300 nm) (*Figure 3c*), corresponding to a nearly 2-fold increase as compared to lowIR. The limits of deep-tissue microscopy also depend on resolution, which is affected by scattering and aberration of the incident excitation beam (*Cheng et al., 2020*). We thus evaluated how the axial resolution depends on imaging depth and excitation process. For 3PE and 4PE processes, resolution remained higher with increasing imaging depth as compared to the resolution achieved by 2PE, which declined steeply beyond 125 µm (*Figure 3d*). Thus, 3PE and 4PE modalities provide improved resolution in strongly scattering tissues compared to 2PE, which may be explained by the superior spatial confinement of the 3P or 4P excitation processes (*Xu et al., 1996*; *Hell et al., 1996*). In addition, light scattering and aberration may affect the quality of the focus and, thereby, resolution in a wavelength-dependent manner (*Cheng et al., 2020*). To address the attenuation of 3PE with increasing tissue penetration in tumors, we measured the fluorescence intensity as a function of increasing imaging depth and derived the effective attenuation length $l_e$ (*Figure 3e*). $l_e$ remained constant over hundreds of micrometers, indicating that the tumor composition was homogenous over this depth range. When red-shifting the excitation wavelength from 1180 to 1300 or 1650 nm, $l_e$ increased from 103 to 128 or 220 µm, respectively. Thus, compared to lowIR excitation, the gain in resolution and SNR in deep-tissue zones with highIR excitation can be attributed to several effects, including: (i) improved localization of the multiphoton effect in the focus region (*Theer and Denk, 2006*; *Xu et al., 1996*), (ii) increased $l_e$ in the 1300 and 1700 nm spectral excitation windows (*Horton et al., 2013*; *Wang et al., 2018b*; *Wang et al., 2018a*), and (iii) improved excitation efficiency as a consequence of increased pulse-energy and low laser repetition rate (*Beaurepaire et al., 2001*; *Theer et al., 2003*). Through these combined effects, highIR excitation increases the imaging depth by 2-fold compared to lowIR excitation using OPO-based sources and by 4-fold compared to lowIR excitation in the 800–900 nm range (*Andresen et al., 2009*).

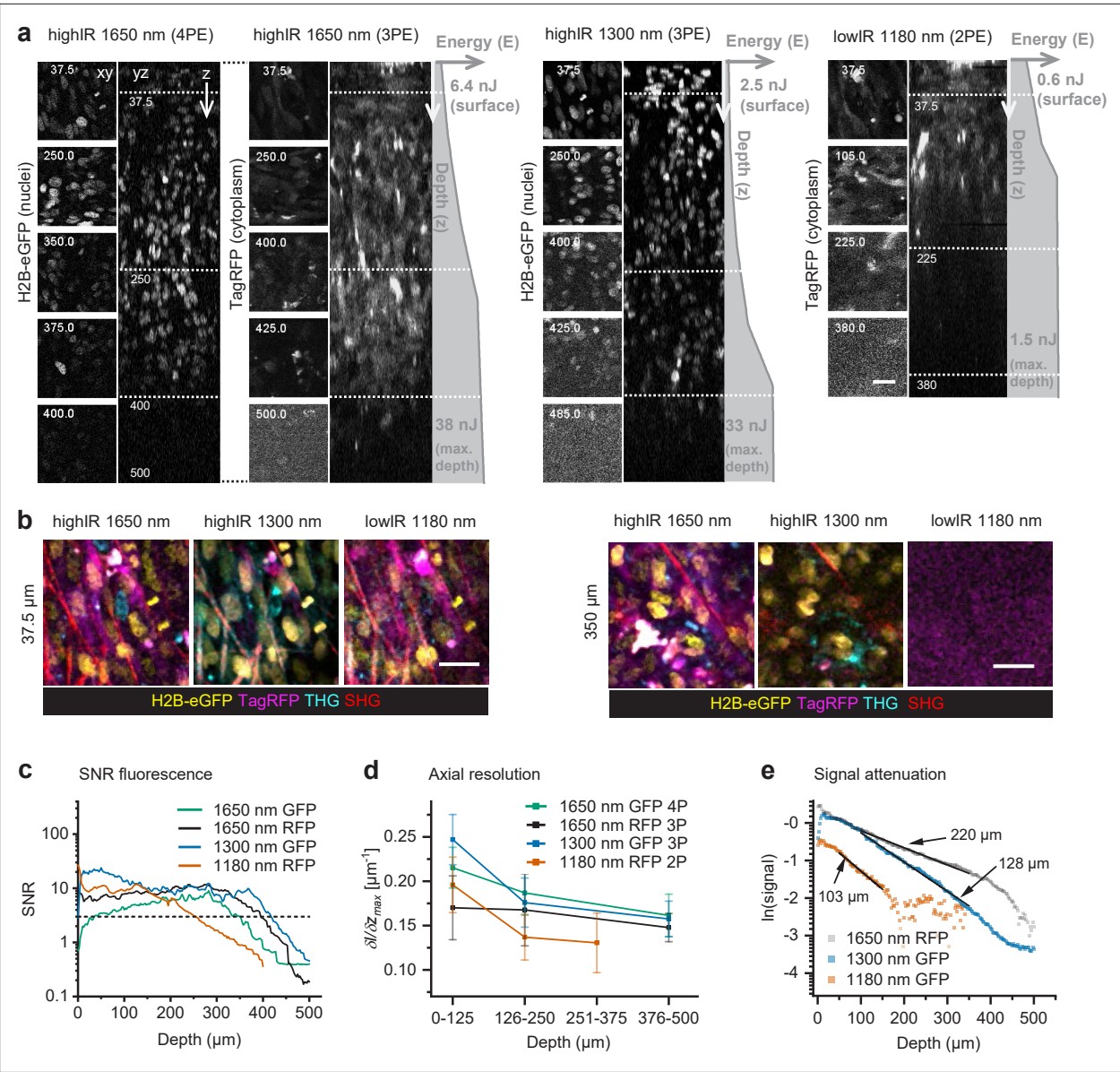

**Figure 3.** Tissue penetration of 2- (2P), 3- (3P), and 4-photon (4P) microscopy in tumors in vivo. After 15 days growth, an intradermal HT-1080 sarcoma tumor expressing eGFP and TagRFP was repetitively imaged with highIR and lowIR excitation to compare tissue penetration of the respective excitation modalities. (**a**) Orthogonal (yz-) views of fluorescent tumor excited at 1650 nm (3PE and 4PE), 1300 nm (3PE), and 1180 nm (2PE). Z-stacks were recorded with increasing excitation pulse energy $E$ at the sample surface with increasing imaging depth $z$ (gray profiles right from orthogonal views). Left, representative (contrast enhanced xy-) images at different depths, represented by the dotted horizontal lines in the yz-views. Details from images: 2.2 s frame time, 20 µs pixel integration time. Numbers: depth in µm. Bar: 25 µm. (**b**) HighIR excitation enables simultaneous multiparameter microscopy up to 350 µm depth, in contrast to lowIR multiparameter microscopy. Images were processed (median filtered, 1 pixel). Bar: 25 µm. (**c**) Signal-to-noise ratio (SNR) of highIR- versus lowIR-excited fluorescent signals as a function of imaging depth, derived from the images shown in (**a**). Horizontal dotted line: detection limit ($SNR = 3$). (**d**) Axial resolution of the fluorescence signals derived for three depth ranges of the images in panel (**a**). The steepness of the transition between the normalized intensity of a fluorescent feature and its nonfluorescent surrounding along the z-direction (($\delta I/\delta z)_{max}$) was taken as a measure for resolution. For each depth range, the median and standard deviation were calculated over 11–17 fluorescent features per channel. (**e**) Attenuation of fluorescence signals for applied excitation wavelengths. The 3P-excited fluorescence intensities derived from (**a**) were normalized to the cubic of the calculated laser power at the sample surface and to the order of the excitation process to obtain normalized signal $S$, and plotted as a function of imaging depth (see *Materials and methods*). For 3PE, the effective attenuation length $l_e$ of the tissue was defined as the depth at which fluorescence intensities attenuate by $1/e^3$, where e is Euler's number (*Wang et al., 2018b*). $l_e$ (indicated as numbers) were derived from single exponential decay functions (black lines) fitted to the normalized data. Source data files: *Figure 3—source data 1* and Figure 3—source data 2.

The online version of this article includes the following source data for figure 3:

*Figure 3 continued on next page*

*Figure 3 continued*

**Source code 1.** Derive relative resolution based on fiji intensity-z-profiles.

**Source data 1.** Numerical data, calculations and templates used to generate graph panels.

## Improved THG imaging depth of highIR over lowIR in soft and dense tissue models

Lastly, we compared lowIR and highIR excitation for THG microscopy in tissues with various scattering properties: tumor, bone, and brain. In the tumor, highIR-excited THG resolved extracellular and cellular details (arrowheads) at depths beyond 400 µm, while lowIR resolved primarily extracellular details until a depth of ~150 µm (*Figure 4a*). SNR analysis showed a maximum imaging depth of >400 µm for highIR and <100 µm for lowIR (*Figure 4b*). Imaging depth limits of THG microscopy with lowIR excitation results from the lower photon density in the focus, leading to a drastically lower excitation efficiency, in combination with the rapid attenuation of nonlinear signals upon scattering and absorption of excitation light (*Beaurepaire et al., 2001*; *Theer et al., 2003*). In contrast, THG by highIR excitation, which creates a multifold higher photon density in the focus, is achieved with only a small fraction of the available power at the tissue surface, which allows to increase the excitation energy at the sample surface with imaging depth further, to maintain a constant signal over a much broader depth range. Interestingly, different details were accentuated in THG images recorded with different excitation regimes. For example, THG by small collagen fibers under non-scattering conditions was best with excitation at 1180 nm, whereas THG by tumor cell nuclei was strongest at 1300 nm excitation (*Figure 4a*, arrowheads). The wavelength-dependent size of the focal volume relative to THG-generating cell and tissue structures is a possible explanation for these differences (*Débarre et al., 2006*). Bone is a strongly light-scattering tissue, yet thin cortical bone such as the mouse skull is amenable to highIR excitation (*Wang et al., 2018a*; *Wang et al., 2019*). To address whether thick bone can be effectively penetrated by highIR, we performed THG microscopy in an excised ossicle, which is an ectopic bone induced by a tissue engineering approach in the mouse (*Dondossola et al., 2018*). Subcellular structures were reliably resolved by highIR excitation beyond 220 µm depth, including osteocyte lacunae and canaliculi in the cortical bone layer and trabeculae in the bone marrow (*Figure 4c*; yz-projections and arrowheads). At comparable pulse energies and near the surface (<105 µm, 1300 versus 1270 nm), a superior SNR was obtained with lowIR excitation, taking advantage of the 80 times higher repetition rate and thus increased emission photon flux (*Figure 4d*). Beyond 150 µm depth, lowIR excitation reached its maximum available power, while highIR excitation power could be further increased to maintain a constant SNR. Thus, as a consequence of the improved excitation efficiency of highIR excitation, superior SNR beyond 165 µm scan depths was obtained compared to lowIR (*Figure 4d*, dotted lines). In addition, 1650 nm highIR excitation resulted in both increased effective attenuation length $l_e$ and larger imaging depth compared to 1270 nm lowIR excitation (*Figure 4e*). Thus, consistent with previous reports in thin bone samples (*Wang et al., 2019*), highIR excitation improves deep-tissue bone microscopy and imaging depth is doubled when using 1300 or 1650 nm highIR excitation compared to 1270 nm lowIR excitation. We finally investigated the imaging depth limit in the mouse brain (*Figure 1—figure supplement 3b*). The SNR remained above the detection limit up to 928 µm depth for 1650 nm highIR, and up to 444 µm for 1270 nm lowIR excitation (*Figure 1—figure supplement 3e*, dotted lines). HighIR imaging depth was ultimately limited by the dense external capsule with short $l_e$, which was not reached with lowIR THG imaging (*Figure 1—figure supplement 3b, d, and e*).

When comparing the applicability of highIR for deep imaging, the depth gain for THG imaging was at least 2-fold compared to lowIR excitation and irrespective of tissue type and density, including brain, tumor, and bone (*Figure 5a*). We experienced stable imaging conditions by carefully increasing the excitation energy with increasing tissue penetration until the maximum energy was reached, followed by rapid deterioration of the signal. For tissues with higher scattering properties (smaller $l_e$), such as bone and cell-rich tumors, the excitation energy E needed to be increased faster with imaging depth, so that the average excitation power reached its maximum value at shallower imaging depths. For 1300 nm highIR excitation in tumor, we used maximum average powers (33 nJ at 1 MHz = 33 mW, *Figure 3a*) well below the reported limits for tissue heating (150 mW) (*Wang et al., 2018a*; *Rowlands et al., 2017*), and therefore the imaging depth can be extended further for this modality. However,

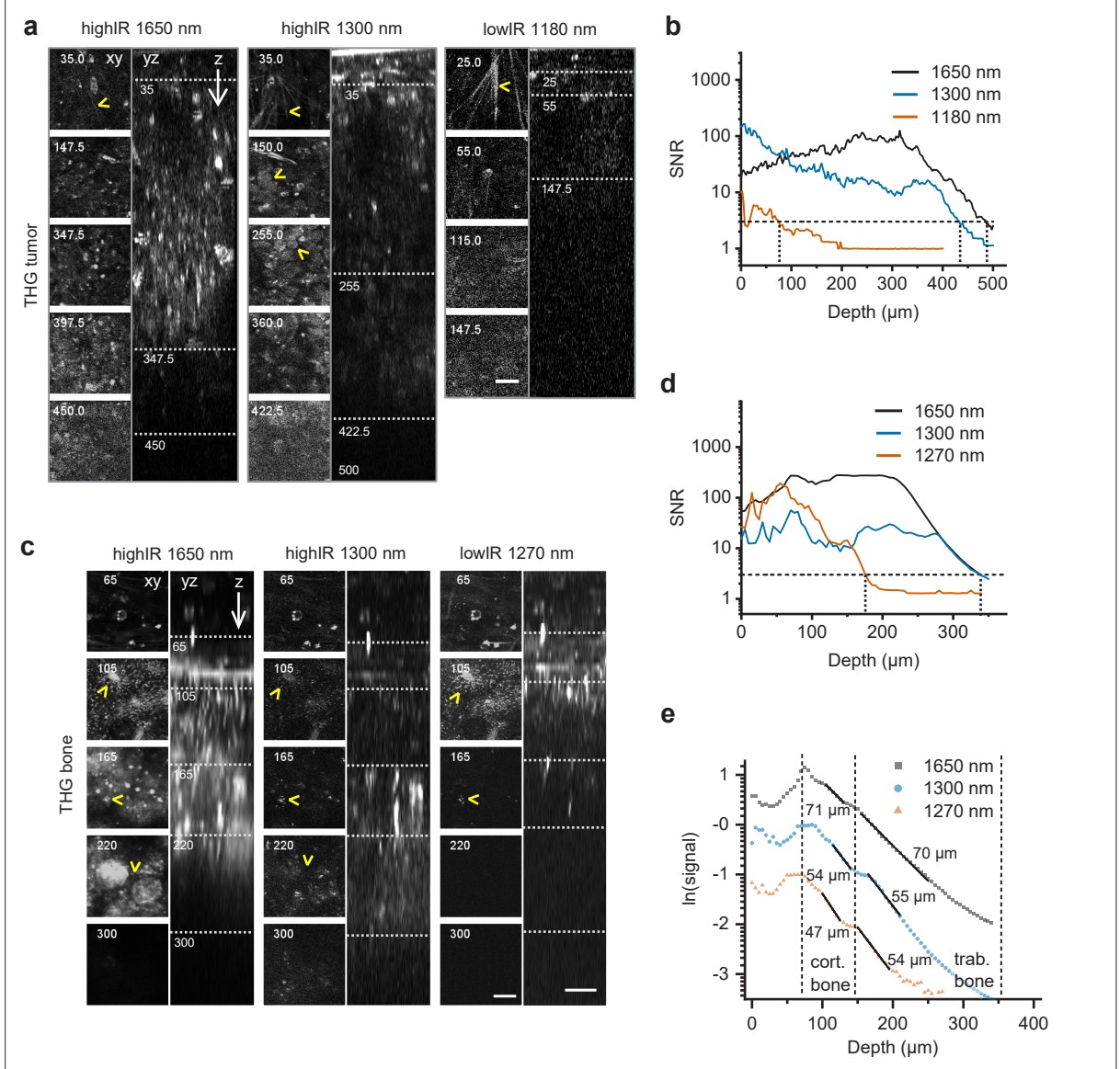

**Figure 4.** Tissue penetration of highIR- and lowIR-excited third harmonic generation (THG) microscopy in tumor and bone. (**a**) THG microscopy in the tumor. THG images were registered simultaneously with the modalities presented in *Figure 3a*. The xy-images represent the intersections in the yz-images (dotted lines). Arrowheads: collagen fibers (top xy-images), tumor cell nuclei (1300 nm xy-images). Numbers: depth in μm. Bar: 25 μm. (**b**) Signal-to-noise ratio (SNR) as a function of imaging depth, derived from the data shown in panel (**a**). Horizontal dotted line: detection limit (*SNR = 3*). Vertical dotted lines: SNR-derived depth limit. (**c**) THG microscopy of ex vivo bone scaffold. Arrowheads: canaliculi (105 μm), channels (165 μm), and sheets of trabecular bone (220 μm). Excitation pulse energy, increasing from sample surface with imaging depth to the maximum: 4.6–32 nJ (left, 1650 nm highIR), 1.3–30 nJ (middle, 1300 nm highIR), and 1.1–1.9 nJ (right, 1270 nm lowIR). Details from images: 2.3 s frame time, 6 μs pixel integration time, 5 μm z-step size. Bars: 25 μm. (**d**) SNR as a function of imaging depth, derived from the image data shown in panel (**c**). Horizontal dotted line: detection limit (*SNR = 3*). Vertical dotted lines: SNR-derived depth limit. (**e**) Attenuation of THG signals from the bone as a function of imaging depth. The signals derived from (**c**) were normalized (see *Materials and methods*) and effective attenuation lengths *l_e* (indicated as numbers) were derived for cortical and trabecular bone layers (black lines). Source data files: *Figure 4—source data 1* and Figure 4—source data 2.

The online version of this article includes the following source data for figure 4:

**Source data 1.** Numerical data, calculations and templates used to generate graph panels.

additional experiments and simulations defining linear phototoxicity limits need to be performed for the 1700 nm excitation range. It remains to be established to what extend the maximum tolerable power depends on differences in water absorption at 1700 nm compared to 1300 nm (*Horton et al., 2013*; *Cheng et al., 2014*). Tissue heating may be reduced and imaging depth increased by reducing

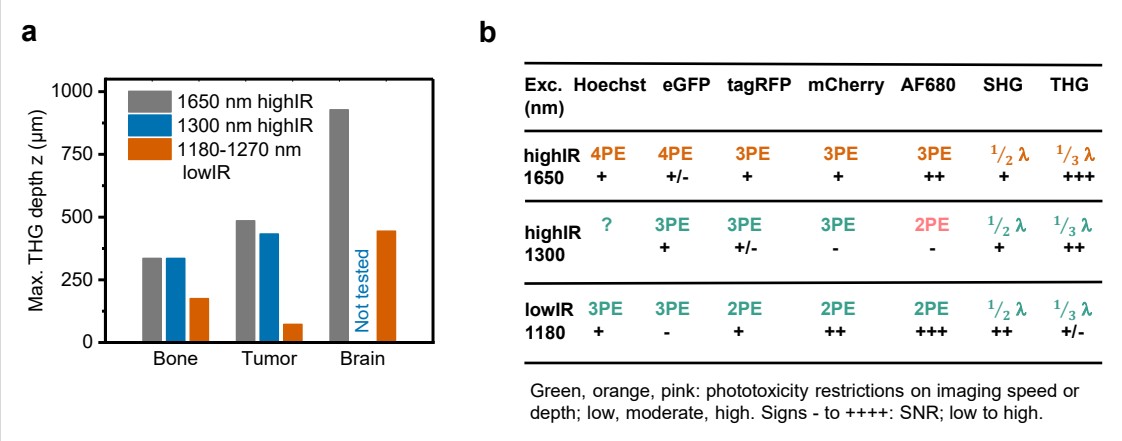

**Figure 5.** Performance comparison of 2P, 3P, and 4P processes excited by highIR versus lowIR. (**a**) Third harmonic generation (THG) imaging depth limit in hard to soft tissues based on the signal-to-noise ratio (SNR) detection limit. Maximum imaging depth was derived from the SNR curves presented in *Figures 3c, 4b and d* and *Figure 1—figure supplement 3d*. (**b**) Summary of applicability of 2PE, 3PE, and 4PE of different fluorophores, based on signal strength and the required restrictions on imaging parameters to avoid phototoxicity. For each fluorophore multiphoton process (2-, 3-, or 4PE), the SNR performance and the severity of restrictions on imaging parameters to avoid phototoxicity are indicated. For THG and SHG, the emission wavelength (fraction of excitation wavelength $\lambda$) and the amount of signal are indicated. Source data files: *Figure 3—source data 1* and *Figure 4— source data 1*.

the imaging frame rate, which allows the tissue to cool down between images or lines, or by reducing the frequency of excitation pulses. This is possible when the biological process of interest can be monitored with lower frame rates or image resolution. For example, studying cancer cell invasion only requires image acquisition with 10 min intervals, which allows effective heat dissipation during image acquisition; conversely, high scan rates required for calcium imaging deeply inside the tumor (20 s) may require a higher repetition rate of excitation pulses and preclude effective deep imaging in highly scattering tissue (*Weigelin et al., 2021*).

We monitored laser-induced toxicity, including nonlinear effects in the focus and linear thermal toxicity in out-of-focus tissue, using a dual toxicity reporter strategy for transient cell stress (GCaMP6 $Ca^{2+}$ reporter) and irreversible membrane damage and cell death (Sytox DNA probe). The $Ca^{2+}$ detection of phototoxic cell stress was faster and more sensitive than recording of irreversible cell death. Additionally, the $Ca^{2+}$ reporter enabled to detect damage which remained sublethal within the time frame of the experiments, which may affect cell functionally but could escape detection by commonly used live-death reporters, such as DNA intercalating probes. Cell stress and bleaching were induced in the focal plane when pulse energies reached threshold values (3.9 nJ for 1300 nm and 12 nJ for 1650 nm at 50 µm depth), eventually leading to physical disruption of tissue structures (burning marks), in line with observations by others (*Hopt and Neher, 2001*). Conversely, energy doses below the thresholds were tolerated by cells without in-focus $Ca^{2+}$ elevation, and this energy level was sufficient to achieve high-quality multiparametric image acquisition at matched depth. The in-focus phototoxicity thresholds obtained in the cell-rich tumor model are comparable to the dynamic range detected in low-density tissues (*Ouzounov et al., 2017*; *Yildirim et al., 2019*; *Wang et al., 2020*), indicating that nonlinear phototoxicity might have a general cause, such as light-induced ionization or formation of reactive oxygen species (*Débarre et al., 2014*; *Tirlapur et al., 2001*), irrespective of the tissue density. The impact of sub-threshold highIR light exposure on long-term integrity of cell structure and function requires further exploration, including growth, differentiation, and chromatin integrity (*Débarre et al., 2014*), as well as benchmarking of stress-sensitive complex cell functions, such as cell migration and cell-cell interactions (*Weigelin et al., 2021*; *Kiepas et al., 2020*).

We generated up to six-channel multiparametric images in live tumors with sufficient signal-to-noise ratio using a single highIR excitation, simultaneously detecting 3PEF and 4PEF along with SHG and THG (*Figure 5b*). To achieve multiparametric imaging, the used excitation pulse energy in the focus was higher compared to previous studies: 1.4–2.1 times for 1650 nm (*Horton et al., 2013*; *Li et al., 2020*; *Guesmi et al., 2018*) and 0.7–1.7 times for 1300 nm (*Ouzounov et al., 2017*; *Wang et al., 2018a*), but still below the defined phototoxicity thresholds. Other methods to achieve multichannel

imaging with highIR excitation have been demonstrated, such as simultaneous 1300 and 1700 nm highIR excitation (*Guesmi et al., 2018*). The method demonstrated here offers the advantage of single laser line excitation and therefore the absence of chromatic aberration and reduced excitation load. On the other hand, the requirements for fluorophore concentration and brightness are stricter for 4P, as compared to 3P excitation processes (*Figure 5b*), due to their even smaller absorption cross-section (*Cheng et al., 2014*). Therefore, fluorophores with high quantum efficiency and/or abundance, such as H2B/eGFP or Hoechst must be selected for 4P excitation. Besides addition of fluorescence channels by combination of 3P and 4P excitation, multiparametric imaging was also achieved by combination of 3P-excited fluorophores with different excitation/emission spectra, such as TagRFP, mCherry, and AlexaFluor680 at 1650 nm or TagRFP and eGFP at 1300 nm. At 1300 nm however, red fluorescent proteins were excited with reduced efficiency. Recently, the excitation efficiency of red fluorescent proteins at 1300 nm was enhanced by carefully tuning the central excitation wavelength to higher-energy electronic excited states, enabling multiparametric imaging with high SNR in three fluorescent channels, at the cost of accelerated photobleaching (*Hontani et al., 2021*). Thus, multi-parametric imaging is a promising approach for the intravital study of complex tissue structures, but labeling and imaging conditions require careful optimization.

Upcoming technical improvements of highIR microscopy include lateral, axial, and temporal multi-plexing (*Weisenburger et al., 2019*), refined compensation of pulse broadening and detection effi-ciency (*Yildirim et al., 2019*) and selective regional scanning (*Li et al., 2020*), providing means to reduce the photon burden and latent phototoxicity to advance imaging speed and depth. Further-more, the advantages of lowIR excitation at shallow depths and highIR excitation at deep imaging regions can be combined, resulting in optimized imaging speed and SNR over the full imaging depth range (*Weisenburger et al., 2019*; *Wang et al., 2020*).

## Conclusion

In conclusion, we find that the benefits of highIR excitation (1300/1700 nm, 1–2 MHz, sub-100 fs pulse length, and nJ-range pulse energy in the focus) previously demonstrated for brain deep-tissue microscopy (*Horton et al., 2013*) also prevail in even more strongly scattering tissues, including thick tumors and bone (*Figure 5a*). HighIR excitation has two important advantages in the context of tumor microscopy: it can roughly double the imaging depth compared to previous state-of-the-art 2P imaging, and it gives access to multiparametric imaging through the simultaneous detection of 3- and 4P fluorescence excitation and multi-harmonic signals (*You et al., 2018b*). We note that using several chromophores simultaneously with a single beam may require to adjust their relative labeling densities and to experimentally determine the best excitation wavelength to optimize the signal and average excitation power threshold to avoid phototoxic effects depending on the application (*Figure 5b*). Based on its simple single-beam excitation and improved imaging depth, highIR microscopy holds great potential to advance biomedicine and material sciences. In cancer research, highIR excitation will improve intravital microscopy of understudied regions, including the tumor core and necrosis zones (*Deryugina and Kiosses, 2017*). Beyond cancer, highIR excitation should advance live imaging of structurally challenging tissues, including the bone marrow (*Kim and Bixel, 2020*), organoids, and embryos (*Débarre et al., 2014*; *Guesmi et al., 2018*). In addition, the much-improved THG signal, together with SHG, 3- and 4P fluorescence excitation of molecular reporters and/or endogenous fluorophores, will allow to record cell type and dynamic functions in a broader morphological context (*Hontani et al., 2021*; *You et al., 2018b*; *Guesmi et al., 2018*), such as cell activities and inflammation in the tumor microenvironment and label-free intra-operative histology (*You et al., 2018b*; *You et al., 2018a*; *Zhang et al., 2019*; *Weigelin et al., 2016*).

## Materials and methods

### Key resources table

| Reagent type (species) or resource | Designation | Source or reference | Identifiers | Additional information |
|---|---|---|---|---|
| Strain, strain background (murine, male) | C57Bl/6J WT | Charles River, Germany | C57Bl/6J WT | 8 to 24 weeks of age |

*Continued on next page*

*Continued*

| Reagent type (species) or resource | Designation | Source or reference | Identifiers | Additional information |
|---|---|---|---|---|
| Strain, strain background (murine, male) | BALB/c CAnN.Cg-Foxn1nu | Charles River, Germany | BALB/c CAnN.Cg-Foxn1nu | 8 to 24 weeks of age |
| Cell line (murine) | B16F10 | ATCC, VA | RRID:CVCL_0159 | |
| Cell line (*Homo sapiens*) | HT1080 | DSMZ, Braunschweig, Germany | RRID:CVCL_0317, ACC315 | |
| Transfected construct (*Rattus norvegicus*) | GCaMP6s | https://doi.org/10.1038/nature12354 | RRID:Addgene_40753 | *Chen et al., 2013* |
| Biological sample (*Rattus*) | Collagen I solution | Corning, NY | REF 354249 | Final concentration 4 mg/ml |
| Chemical compound, drug | Dextran70-AlexaFluor680 | Invitrogen, CA | C29808 | 20–100 µl, 20 mg/ml in saline |
| Chemical compound, drug | Hoechst 33342 | Sigma-Aldrich | 14533 | 1.1 mg in milliQ |
| Chemical compound, drug | Sytox Green | Thermo Fisher, Waltham, MA | S7020 | 5 mM |
| Software, algorithm | Origin 2019 | OriginLab Corporation, MA | RRID:SCR_002815 | https://www.originlab.com/viewer/ |
| Software, algorithm | Fiji/ImageJ | https://doi.org/10.1038/nmeth.2019 | version 1.52n | https://imagej.net/software/fiji/ *Schindelin et al., 2012* |
| Software, algorithm | Matlab | Mathworks | R2017b | https://www.mathworks.com |
| Other | 25 × 1.05 NA water immersion objective lens | Olympus, Tokyo, Japan | XLPLN25XWMP2 | |
| Other | 0.2 µm multicolor beads | Polysciences Inc, PA | Cat. 24050 | FluoresBrite 0.2 µm |

## Imaging setup

The setup was based on a customized upright multiphoton microscope (TrimScope II, LaVision BioTec, a Miltenyi Biotec company, Bielefeld, Germany) equipped with two tunable Ti:Sa lasers (Chameleon Ultra I and II, Coherent, CA), an OPO (MPX, APE, Berlin, Germany) and up to six PMTs distributed over a two- and a four-channel port (*Figure 1—figure supplement 1a*). The setup was modified to facilitate high-energy, low-repetition rate excitation. A high-power fiber laser (Satsuma HP2, 1030 nm, 20 W, 1 MHz Amplitude Systèmes, Bordeaux, France) was used to pump an OPA (AVUS SP, APE), which generated 460 or 330 mW at 1300 or 1650 nm, respectively. A fixed-distance prism compressor (Femtocontrol, APE), glass block (IR-coated 25 mm ZnSe, for 1650 nm), and an autocorrelator with internal and external detector (Carpe, APE) were included in the optical path to control the pulse length under the objective lens. The beam path further included an adjustable 2:1 telescope (*f* = 80 mm and *f* = 40 mm apochromat lenses; IR-coated), a motorized half-wave plate and a glan-laser polarizer to control laser beam diameter, power, and polarization. The pulse length under the objective lens and its point-spread-function were optimized for the chosen OPA excitation wavelength by adjustment of the excitation path bulk compression, beam pointing, and telescope, such that the objective lens' back focal plane was 10% overfilled. A movable mirror was used to guide either the OPO or the OPA beam into the scanhead, where it was spatially overlaid with the Ti:Sa beam. Mirrors, dichroic mirrors, and lenses in the scanhead were carefully selected for high reflectance or transmission in the extended excitation wavelength range. Microscopy was performed using a 25 × 1.05 NA water immersion objective lens (XLPLN25XWMP2, Olympus, Tokyo, Japan; transmission of 69% at 1700 nm, data not shown). The following filter/PMT configurations were used: blue-green emission was split off to a two-channel port with a 560lp dichroic mirror and a 700SP laser blocker filter, while red emission was split off to a two-channel port with a 900lp dichroic mirror and an 880SP laser blocker filter. Red emission was first split by a 697sp, then further split by a 605lp and a 750sp dichroic mirror, bandpass filtered with 572/28 (TagRFP) or 593/40 (TagRFP and mCherry), 620/60 (mCherry), 710/75 (AlexaFluor680), and 810/90 (AlexaFluor750, SHG) and detected by alkali, GaAsP, or GaAs PMT detectors (H6780-20, H7422A-40, or H7422A-50, Hamamatsu, Hamamatsu city, Japan). For 1180, 1270, and 1300 nm excitation, blue-green emission was split by a 506lp dichroic mirror, bandpass filtered with 447/60 (THG) and 525/50 (eGFP) and detected by alkali or GaAsP detectors (H6780-01, H6780-20, or H7422A-40, Hamamatsu).

For 1650 nm excitation, blue-green emission was split by a 506lp (more THG signal) or 560lp (more eGFP signal) dichroic mirror, bandpass filtered with 447/60 (Hoechst) or 505/40 (eGFP) and 562/40 (THG) and detected by alkali or GaAsP detectors (Hamamatsu, H6780-01, H6780-20, or H7422A-40). Filters were fabricated by Semrock (New York, NY) or Chroma Technology GmbH (Olching, Germany). The setup was equipped with a warm plate (DC60 and THO 60–16, Linkam Scientific Instruments Ltd, Tadworth, UK) and a custom-made objective heater (37°C) for live-cell and in vivo experiments, as described (*Haeger et al., 2020*).

## Determination of setup resolution

The point-spread-function was obtained using 0.2 µm multicolor beads (FluoresBrite 0.2 um, Cat. 24050, Polysciences Inc, PA). Beads were washed, suspended in agarose (A4718, 1 %w in 1× phosphate buffered saline (PBS), Sigma-Aldrich, MO), and scanned through a coverglass (18 × 18 mm$^2$ #1, Menzel-Glaeser, Braunschweig, Germany). Z-stacks of 30 µm depth were recorded with 0.5 µm step interval, 0.24 µm pixel size, 1.0–1.5 µs pixel dwell time, and 4- to 10-fold line averaging and with 1650 nm (5.5 nJ, sample surface), 1300 nm (3.5 nJ, sample surface), 1280 nm (89 mW, under objective), or 910 nm (13 mW, under objective) excitation. Emission was bandpass filtered with 562/40 (1650, 1300, and 910 nm) or 620/60 (1280 nm) and detected with a GaAsP detector (specified above). The software PSFj (*Theer et al., 2014*) was used for point-spread-function analysis.

## Intravital imaging procedures

Intravital microscopy of intradermal tumors was performed as described (*Haeger et al., 2020*). In brief, the animal was anesthetized (1–2% isoflurane in $O_2$ for up to 4 hr), and vessels visualized using intravenously injected Dextran70-AlexaFluor680 (25 min prior to multiphoton imaging, 20–100 µl, 20 mg/ml in saline, C29808, Invitrogen, CA). At endpoint sessions, Hoechst 33342 (14533, 1.1 mg in milliQ, Sigma-Aldrich) was injected intravenously to visualize cell nuclei. To define regions of interest, overview images were obtained using an Olympus XL Fluor 4×/340 objective lens and epifluorescence excitation (X-Cite 120 lamp, Excelitas, MA; Olympus GFP/RFP filter block and a 2/3″ cooled CCD camera) (*Figure 1—figure supplement 1e*). Prior to multiphoton imaging, the maximum average power under the objective was measured (FieldMaxII-TO power meter with PM2 sensor, resolution 1 mW, Coherent) and the excitation energy at the surface of the sample (*Figure 1—figure supplement 2*) was adjusted below the found functional toxicity threshold (*Figure 2*, *Figure 2—figure supplement 1*). To maintain a constant 3P emission over imaging depth, the excitation power was increased as defined in the measurement. For image acquisition, the pixel dwell time was set to 2 or 4 µs to synchronize with the laser repetition rate, line averaging was set between 1 and 6, pixel size was 0.7 µm and the step size of z-stacks was 2.5 µm unless stated otherwise. To ensure maximum compatibility of highIR excitation with open-window longitudinal imaging of in vivo tumors models, saline was used as immersion liquid.

## Brain imaging ex vivo

At the endpoint, a tumor-bearing 9-week-old C57BL/6J mouse was anesthetized, intravenously injected with Dextran70-AF680, and sacrificed. The brain was excised, placed in a PBS filled container, and covered with a #1 microscope coverglass (Menzel-Glaser). Z-stack images were acquired in the neocortex above the hippocampus area, with 12 µs pixel dwell time, pixel size 0.50 µm, and 4 µm z-step size. Multiple measurements were performed, to optimize either THG and/or AF680 emission for different depth ranges, for 1650 and 1270 nm excitation wavelengths (*Supplementary file 1c*). Measurements were combined to generate signal attenuation curves and to compose one image stack with maximized penetration depth.

## Image processing, data representation, and statistics

Unless stated otherwise, image processing was performed with Fiji/ImageJ, version 1.52n (*Schindelin et al., 2012*). Part of the datasets contained positional jitter, which was removed with the Image Stabilizer plugin (*Li, 2008*). Unless stated otherwise, Origin 2019 (OriginLab Corporation, MA) was used for numerical and statistical calculations, data fitting, and representation.

## Study of the multiphoton processes underlying multimodal excitation

Excitation power under the objective was calibrated for all used attenuator settings. Images were acquired with stepwise decreasing– increasing excitation power (*P*). For bleaching correction,

reference images were taken after each image at a fixed low excitation power ($P_{bleach}$). All images were acquired in one imaging plane, with pixel size 0.74 μm and pixel integration time 6.0 μs. For each channel, individual images were merged into two stacks; one for excitation power and one for bleaching correction. To quantify intensities, bright pixels and background pixels were selected by gating with a manually drawn region of interest (ROI) and/or by multiplication of the image stack with a binary mask. Masks were created by a combination of median filtering, auto-thresholding, and binary erode steps. For the selected pixels, area, mean intensity, and standard deviation (resp. $I_{mean}$, $\sigma_I$ for bright pixels; $B_{mean}$, $\sigma_B$ for background) were quantified along the excitation and bleaching correction stacks. Normalized, bleaching corrected, background subtracted mean intensity ($S$) in relation to excitation power was derived as follows: $S(P) = F_{norm} \cdot [I_{mean}(P) - B_{mean}(P)]/[I_{mean}(P_{bleach}) - B_{mean}(P_{bleach})]$, where $F_{norm}$ is a constant for normalization. To estimate the order of the excitation process ($n$), a power function $S(P) = A \cdot P^n$ was fitted to the data, with $A$ the proportional factor. The orthogonal distance regression iteration algorithm was applied to include both $P$ (measurement inaccuracy) and $S$ (linear approximation including pixel noise and normalization) errors in the fitting process. Reduced chi-square and adjusted R-square values were below 2 and above 0.995, respectively. Standard errors were given for $A$ and $n$.

## Analysis of fluorescence bleaching

The H2B channel (1300 nm, eGFP or 1650 nm, mCherry) of the 3D + time stack was mean (2) filtered and average projected over the z-axis. Bright pixels in cells were selected by auto-thresholding (1300 nm, Huang or 1650 nm, Iso) in combination with manual selection of an ROI and their average intensity was obtained. The average background was calculated over the manually selected darkest region of the image stack and subtracted from the cell-based fluorescence signal for every time point, to obtain the background subtracted fluorescence signal as a function of time.

## SNR analysis

The SNR as a function of imaging depth was calculated for every position in the depth stack from the average fluorescence intensity ($I_{mean}$) over the brightest 1st (THG signal), 10th (nuclei, eGFP), or 40th (cytosol, TagRFP) percentile of pixels in the median filtered (2, fluorescence; 1, THG) image. As background signal, the average ($B_{mean}$) and standard deviation ($\sigma_B$) were calculated over an ROI in a dark location of the unfiltered stack. Then, the SNR was calculated as $SNR = (I_{mean} - B_{mean})/\sigma_B$. The SNR along a line profile was obtained from the intensity values along the line ($I_{line}$), as $SNR = (I_{line} - B_{mean})/\sigma_B$.

## Axial resolution analysis of in vivo data stacks

Pixels in fluorescent cell bodies or nuclei in the z-stack were selected with a fixed-size ROI, their average intensities were calculated over the ROI and intensity z-profiles were generated. Intensity z-profiles were normalized (0–1) and their maximum derivatives (($\delta I/\delta z)_{max}$) were calculated (*Figure 3—source code 1*, custom script, Matlab). Median and standard deviation values were derived over sets of ($\delta I/\delta z)_{max}$ and were used as a relative measure for axial resolution.

## Signal attenuation analysis

To display signal attenuation curves independent of excitation power at the sample surface ($P$) and independent of the order of the excitation process ($n$), average pixel intensities ($I_{mean}$) were background subtracted and normalized to $P$ and $n$, to obtain the normalized signal $S$ for each imaging depth: $S = N \cdot [(I_{mean} - B_{mean})/P^n]^{1/n}$, where $N$ is a normalization constant and $B_{mean}$ is the background. $I_{mean}$ was quantified (described in the paragraph *SNR analysis*) and $B_{mean}$ was estimated by averaging all the pixel values of the last frame of the image stack. For 3P processes, the effective attenuation length $l_e$ of the tissue was defined as the depth at which $I_{mean}$ attenuates by $1/e^3$, where $e$ is Euler's number and $I_{mean}$ the average intensity of the 3P process (*Horton et al., 2013*). To obtain $l_e$, the normalized signal $S$ of the attenuation curves was fitted with a single exponential function $S(z) = A \cdot exp(-z/l_e)$, where $A$ is a proportional constant and $z$ the imaging depth. Note that for 3P processes ($n = 3$), the signal $S$ shown in the attenuation curves was normalized as: $S(z) \propto (I_{mean})^{1/3}$. Therefore, fitting the normalized signal $S(z)$ with $A \cdot exp(-z/l_e)$ is similar to fitting $(I_{mean} - B_{mean})/P^3$ with $A \cdot exp(-3z/l_e)$.

## Cells and cell culture

Murine B16F10 melanoma cells (ATCC, VA) were cultured in RPMI (Gibco) supplemented with 10% FCS (Sigma-Aldrich), 1% sodium pyruvate (11360, GIBCO, MA), and 1% penicillin and streptomycin

(PAA, P11/010) at 37°C in a humidified 5% $CO_2$ atmosphere. Human HT1080 (ACC315) fibrosarcoma cells (DSMZ, Braunschweig, Germany) were cultured in DMEM (Gibco) supplemented with 10% FCS (Sigma-Aldrich), 1% sodium pyruvate (11360, Gibco), and 1% penicillin and streptomycin (PAA, P11/010) at 37°C in a humidified 5% $CO_2$ atmosphere. Cell line identity was verified by a SNP_ID Assay (Sequenom, MassArray System, Characterized Cell Line Core facility, MD Anderson Cancer Center, Houston, TX). Cells were routinely tested for mycoplasma using MycoAltert Mycoplasma Detection Kit (Lonza, Basel, Switzerland). HT1080 cells were lentivirally transduced to stably express the fluorescent proteins eGFP or mCherry tagged to histone 2B and cytoplasmic TagRFP. B16F10 cells were lentivirally transduced to stably express the green fluorescent intracellular calcium sensor GCaMP6s (*Chen et al., 2013*) and mCherry tagged to histone 2B.

### 3D spheroid culture

3D spheroid culture was established as described (*Veelken et al., 2017*). Shortly, HT1080 fibrosarcoma cells from sub-confluent culture were detached with 2 mM EDTA (1 mM) and spheroids containing 1000 cells were formed with the hanging drop method. Aggregated spheroids were embedded into a collagen I solution (non-pepsinized rat-tail collagen type I, final concentration 4 mg/ml, REF 354249, Corning, NY) and transferred into a chambered coverglass prior to polymerization at 37°C. After polymerization, chambers were filled with culture medium (specified above), incubated overnight at 37°C in a humidified 5% $CO_2$ atmosphere and sealed prior to microscopy.

### Animal procedures

All animal procedures were approved by the ethical committee on animal experimentation (RU-DEC 2014–031) or the Central Authority for Scientific Procedures on Animals (CCD: AVD10300 2017 4444, license: 2017–0042). Handlings were performed at the central animal facility (CDL) of the Radboud University, Nijmegen, in accordance with the Dutch Animal experimentation act and the European FELASA protocol. C57Bl/6J WT mice and BALB/c CAnN.Cg-Foxn1nu were purchased from Charles River, Germany. Before the experiment, mice were housed in IVCM cages at standard housing conditions. Food and water were accessible ad libitum. Dorsal skinfold chambers (DSFC) were transplanted on 8- to 24-week-old male mice as described (*Haeger et al., 2020*). In short, mice were anesthetized using isoflurane anesthesia (2% in oxygen), the chamber was mounted on the dorsal skinfold of the mice, one side was surgically removed, and a coverglass was used to close the imaging window. Mice received an adequate peri-surgical analgesia using carprofen and buprenorphine. To prevent dislocation and inflammation of the DSFC, mice were housed with reduced cage enrichment during the experiment. Mice were housed in a temperature-controlled incubator at 28°C to minimize heat loss in the skin tissue due to the DSFC. One day after surgery, B16F10 melanoma ($0.5 \times 10^5$) or HT1080 fibrosarcoma ($2 \times 10^6$) were implanted as single-cell suspension into the deep dermis of the mouse using a 30-G needle (one or two tumors per mouse). To monitor tumor progression, mice were briefly anesthetized using isoflurane and epifluorescence overview images were taken (*Figure 1—figure supplement 1e*).

### 3D sample preparation for cell viability experiments in vitro

A chambered coverglass was filled halfway with collagen I solution (4 mg/ml, as described above) and allowed to polymerize (37°C, 5% $CO_2$, humidified atmosphere) in a vertical position. Then, B16F10 melanoma cells from sub-confluent (25%) culture were detached with 2 mM EDTA (1 mM) and resuspended at high density ($\sim 1 \times 10^8$). Approximately 10 µl of cell suspension was transferred into a slightly bended 27-G hypodermic needle (BD) and positioned onto a P20 pipette. The needle was brought into the collagen gel, parallel to the coverglass. A small volume (~1 µl) was injected while retracting the needle to form a rod-shaped cell structure. Spill-over cells were removed by gently washing with PBS. Chambers were filled up with culture medium, placed in a 15 cm culture dish with wet paper to prevent evaporation, and incubated overnight (37°C, 5% $CO_2$, humidified atmosphere). Prior to microscopy (~1 hr), 1 µM of Sytox Green (S7020, 5 mM, LOT #2204228, Thermo Fisher, Waltham, MA) was added to the medium and the sample was sealed.

### Imaging setup for in vitro cell viability measurements

A custom-built setup was used for in vitro cell viability measurements. Shortly, the microscope was based on a lab-built upright frame in combination with the 25 × 1.05 NA water immersion objective,

and a prototype highIR source (LOB and Amplitude Systèmes, Bordeaux, France) was used for excitation. The setup had the following highIR excitation specifications: 1700 nm central wavelength, 78 fs pulse length and >100 mW under the objective, 1 MHz pulse repetition rate, and 3 µm axial resolution for 3P excitation. Thus, these conditions matched those used on the intravital imaging setup. GCaMP6 and Sytox Green were 2P excited with an Axon920 source (Coherent Inc 920 nm, 140 fs, 80 MHz), which focal plane was shifted ~6 µm deeper into the sample as compared to 1700 nm excitation. Scanning and acquisition was controlled using ScanImage software (Vidrio Technologies). Emission was split by a R561 dichroic mirror with the slope centered at 575 nm, bandpass filtered with ET505/40 (Sytox Green and GCaMP6) and 650/100 (mCherry and Sytox Green) and detected by GaAsP detectors (H7422-40, Hamamatsu, Hamamatsu city, Japan). Sample temperature was kept at 37°C using a heating chamber for upright microscopes (H310 UP with an UP-1 × 35 M insert, Okolab SRL, Pozzuoli, Italy) and an objective heater (as described for the intravital setup).

## Cell viability analysis based on tracking of individual cells

In the last recorded composite image of the time-lapse stack (*Figure 2b* most right image), a group of Sytox-positive and Sytox-negative cells were selected by visual inspection. The frame order of the stack was reversed to facilitate manual tracking, which was performed on the nucleus and cytosol of each selected cell (*Figure 2—figure supplement 1a*). Then, a Matlab script (*Figure 2—source code 1*, *Figure 2—source code 2*) was used to retrieve intensity-time traces at the coordinates of the manually tracked cells, from the spatially filtered (multiplication by 10, two-pixel mean) green channel of the raw time-lapse stack. Nucleus and cytosol intensity-time traces of Sytox-positive and Sytox-negative cells were imported into Origin2019 for further processing: the normalized single-cell GCaMP intensity-time traces (*Figure 2—figure supplement 1b*) were obtained by subsequent dark background subtraction, smoothing (method: Savitzky-Golay, window: 10 points, polynomal order: 2) and normalization to maximum of cytosol intensity-time traces. To obtain Sytox intensity-time traces without GCaMP crosstalk, the nucleus signal was divided by the cytosol signal, after background subtraction and prior to smoothing. The other processing steps were similar compared to the generation of GCaMP time traces. Cells were considered responsive (positive) if the trace intensity crossed a value *threshold = mean$_{baseline}$ + 3\* SD$_{baseline}$*, where *mean$_{baseline}$* and *SD$_{baseline}$* are respectively the mean and the standard deviation of the unsmoothed intensity-time trace before highIR exposure. Assuming a normal distribution of trace intensities around *mean$_{baseline}$*, a null hypothesis stating that intensity is below the threshold value can be rejected with $p = 0.0015$, if intensity values are above a threshold value equal to 3\* *SD$_{baseline}$* above the *mean$_{baseline}$* (one-tailed z-test). The Sytox and GCaMP responses over time presented in *Figure 2* were derived by averaging all raw intensity-time traces per group (nucleus and cytosol, Sytox positive and negative) prior to background subtraction and calculation of the ratio nucleus/cytosol for the Sytox response.

## Image area-based cell viability analysis

The cell fraction responding upon highIR exposure (GCaMP+ or Sytox+ cell fraction) was derived from time-lapse and z-stack measurements within an ROI of the image for which cell mass above the highIR focal plane was 40±10 µm thick, unless stated otherwise (*Figure 2a* and *Figure 2—figure supplement 1c*). Here, the green channel images were used for quantification of GCaMP while the red channel was used for Sytox signals. An ImageJ script (*Figure 2—source code 3*, *Figure 2—source code 4*) was used to derive the fraction, in the following manner: first, to discriminate between responding and not responding cell area within frames, a threshold $t = mean + 3*SD$ was defined, where mean and standard deviation (*SD*) were derived over the 95% dimmest pixels within the ROI, drawn in a time-lapse frame taken before highIR exposure. Second, a mask was created for every time point or depth in the median filtered (one-pixel) stack, where pixels above threshold *t* within the ROI are classified as high. In this mask all GCaMP+ or Sytox+ classified pixels were included, also those which were positive prior to highIR exposure. Since the aim was to quantify the positive cell fraction *upon highIR exposure*, the positively classified pixels prior to highIR exposure (caused by spontaneous fluctuations, dead or dying cells, melanin artifacts) were removed from the mask in the next steps. Third, a mask was created from the last time point or from the complete z-stack before highIR exposure, following the steps described above. Additionally, six dilation steps were applied to this mask, to correct for drift and/or expansion of detected features. Fourth, the last mask was subtracted from

the uncorrected mask and the area of the highIR-induced GCaMP+ or Sytox+ pixels was quantified for each frame of the newly created mask. Finally, the area of the ROI corrected for positively classified pixels prior to highIR exposure was measured. The quantified areas per frame were exported to Origin2019 in order to derive the relative GCaMP+ or Sytox+ cell fractions upon highIR exposure for further representation and statistical analysis of time-lapse and z-stack measurements. For time-lapse measurements, the highIR-induced GCaMP+ cell fraction was defined as the maximum fraction obtained during highIR exposure.

## Acknowledgements

We acknowledge Eleonora Dondossola for supplying bone samples; Esther Wagena, Bianca Lemmers-Van de Weem, and Mike Peters for expert technical support and assistance in animal experiments; and Mirjam Zegers for critical reading of the manuscript. We thank Amplitude Systèmes for providing a Satsuma HP2 demo system and Lucie Desclaux, Yoann Zaouter, Alexandre Thai, and Aurelia Durand for hardware support, and we further thank APE GmbH, Berlin, for providing the AVUS SP demo system. Lastly, we gratefully acknowledge Chris Xu, Willy Supatto, Pierre Mahou, Hugo Blanc, Xavier Solinas, Frédéric Druon, Raluca Niesner, Asylkhan Rakhymzhan, and Rafael Kurtz for insightful discussions. This work was supported by the European Research Council (617430-DEEPINSIGHT) to PF, Agence Nationale pour la Recherche (ANR-11-EQPX-0029, ANR-15-CE13-0015) to EB, Fondation pour la Recherche Médicale (DEI201512440) to EB, and the Cancer Genomics Center (CGC.nl) to PF.

## Additional information

### Competing interests

Judith Heidelin: Judith Heidelin is currently employed by LaVision BioTec GmbH and explores implementation of high-pulse-energy low-duty-cycle light sources as a microscopy product line. Volker Andresen: Volker Andresen is currently employed by LaVision BioTec GmbH and explores implementation of high-pulse-energy low-duty-cycle light sources as a microscopy product line. Marcus Beutler: Marcus Beutler has a current employment at APE Angewandte Physik & Elektronik GmbH, which produces the AVUS SP as a commercial product. The other authors declare that no competing interests exist.

### Funding

| Funder | Grant reference number | Author |
| --- | --- | --- |
| FP7 Ideas: European Research Council | 617430-DEEPINSIGHT | Peter Friedl |
| Nederlandse Organisatie voor Wetenschappelijk Onderzoek | NWO Gravitation Programme 024.001.028 | Peter Friedl |
| Agence Nationale de la Recherche | ANR-11-EQPX-0029 | Emmanuel Beaurepaire |
| Agence Nationale de la Recherche | ANR-15-CE13-0015 | Emmanuel Beaurepaire |
| Fondation pour la Recherche Médicale | DEI201512440 | Emmanuel Beaurepaire |

The funders had no role in study design, data collection and interpretation, or the decision to submit the work for publication.

### Author contributions

Gert-Jan Bakker, Conceptualization, Formal analysis, Investigation, Methodology, Project administration, Resources, Software, Visualization, Writing – original draft, Writing – review and editing; Sarah Weischer, Investigation, Resources, Writing – review and editing; Júlia Ferrer Ortas, Conceptualization, Investigation, Methodology, Resources, Writing – review and editing; Judith Heidelin, Marcus

Beutler, Conceptualization, Methodology, Resources, Writing – review and editing; Volker Andresen, Conceptualization, Methodology, Writing – review and editing; Emmanuel Beaurepaire, Conceptualization, Funding acquisition, Investigation, Methodology, Resources, Writing – review and editing; Peter Friedl, Conceptualization, Funding acquisition, Methodology, Supervision, Writing – review and editing

### Author ORCIDs
Gert-Jan Bakker ⬚ http://orcid.org/0000-0003-3602-0014
Peter Friedl ⬚ http://orcid.org/0000-0002-0119-4041

### Ethics
All animal procedures were approved by the ethical committee on animal experimentation (RU-DEC 2014-031) or the Central Authority for Scientific Procedures on Animals (CCD: AVD10300 2017 4444, license: 2017-0042). Handlings were performed at the central animal facility (CDL) of the Radboud University, Nijmegen, in accordance with the Dutch Animal experimentation act and the European FELASA protocol. Before the experiment, mice were housed in IVCM cages at standard housing conditions. Food and water were accessible ad libitum. All surgery was performed using isoflurane anesthesia (2 % in oxygen). Mice received an adequate peri-surgical analgesia using carprofen and buprenorphine. To prevent dislocation and inflammation of the dorsal skinfold imaging chamber, mice were housed with reduced cage enrichment during the experiment.

### Decision letter and Author response
Decision letter https://doi.org/10.7554/eLife.63776.sa1
Author response https://doi.org/10.7554/eLife.63776.sa2

---

## Additional files

### Supplementary files
• Supplementary file 1. Supplementary tables.
(**a**) Comparison of parameters related to linear, lowIR, and highIR excitation modalities. The listed parameters are derived from experiments described in this paper, unless stated otherwise. *Peak power focus* is defined as the maximum excitation power in the focus during the laser pulse. *Power surface* is defined as the average excitation power at the sample surface calculated from the power under the objective and the water absorption (*Figure 1—figure supplement 2*). *Emission I(E)*: dependence of emission intensity on excitation energy. *Attenuation of light, confocal*: very high, both excitation and detected emission (only ballistic photons from the focus pass the pinhole in the emission path) are strongly attenuated by the relatively short effective attenuation length for visible-range excitation (*Helmchen and Denk, 2005*; *Benninger and Piston, 2013*). *LowIR*: less attenuation as compared to visible excitation wavelengths; non-de-scanned detection improves emission detection (*Centonze and White, 1998*). *HighIR:* equal to less attenuation as compared to lowIR excitation. *Water absorption*: contribution of water absorption to the attenuation of excitation light (*Qiu et al., 2017*; *Nachabé et al., 2010*; *Wang et al., 2016*). (**b**) Order of the excitation processes (*n*) from emission intensity as a function of excitation energy (related to *Figure 1*). Data were derived from datasets described in *Figure 1* and *Figure 1—figure supplement 4*. Additional independent measurements obtained by live imaging of multicellular HT1080 spheroids in rattail collagen included 1300 nm highIR excitation of eGFP, TagRFP, SHG, and THG and 1650 nm highIR excitation of TagRFP and mCherry at 0.5 and 1 MHz repetition rates. For fitting of data with 1300 nm excitation, the threshold for physical damage was 16 nJ and the saturation limit was 3.1 nJ for eGFP and TagRFP. Source data files: *Figure 1—source data 1*, Figure 1—source data 2, *Source data 1* and Source data 2. (**c**) Experimental parameters for brain measurements optimized for THG and/or AF680 emission. Data were obtained for 1650 highIR or 1270 nm lowIR excitation and different depth ranges. Line averaging (Line av.), two- or four-channel emission port (Port), dichroic mirror to split off emission (DM), emission bandpass filter (BP), detector type (PMT), and immersion liquid (Imm.). Rows in the table with the same sequence number (Seq.) were acquired simultaneously.

• Transparent reporting form

• Source data 1. Raw microscopy image files, intermediate and analysis results.

## Data availability

All data generated or analysed during this study are included in the manuscript and supporting files. Source data files, including raw image data, have been provided for all figures and tables.

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
