## [Editor Report]

Nonlinear microscopy is in the unique position that high-resolution images of cells and other tissue components can be obtained in live tissue. However, scattering and absorption limit the penetration depth and thus the level of 3D information that can be achieved. In this manuscript, the authors develop a new approach that can accomplish deeper imaging in complex specimens, which has potential to provide new insight into biological mechanisms within living tissues and whole organisms.

---

## [Decision Letter]

**Decision letter after peer review:**

Thank you for submitting your article "Intravital Deep-Tumor Single-Beam 2-, 3-and 4-Photon Microscopy" for consideration by *eLife*. Your article has been reviewed by 3 peer reviewers, and the evaluation has been overseen by a Reviewing Editor and Anna Akhmanova as the Senior Editor. Two of the following individuals involved in review of your submission have agreed to reveal their identity: Scott E Fraser (Reviewer #1); Marloes Groot (Reviewer #2).

The reviewers have discussed the reviews with one another and the Reviewing Editor has drafted this decision to help you prepare a revised submission.

Summary:

Nonlinear microscopy is in the unique position that high-resolution images of cells and other tissue components can be obtained in live tissue. However, scattering and absorption limit the penetration depth. The impact of nonlinear microscopy in biomedicine and biology would be much improved if higher imaging depths can be achieved. In this manuscript, the authors show they can accomplish imaging in complex specimens using 3- and 4-photon excitation, deeper in the specimen than comparable optics can accomplish with 2-photon excitation laser scanning microscopy. Using a customised commercial system, the authors have incorporated a high-powered laser source with an OPA and dispersion compensation to generate either 1330nm or 1650nm laser lines with high peak pulse energies at low pulse repetition rates. They then compare the relative capabilities of each laser line in terms of number of fluorescence emission channels measured (skin tumour xenografts), fluorescence bleaching analysis and functional toxicity thresholds and fluorescence signal attenuation (excised murine bone).

This is a very interesting study with some potentially important findings from a technical perspective. However, there is a disconnect at present between the quality of the work and the quality of the presentation. There are many areas of quantitative imaging and intravital imaging that are well known to those in the direct field, and that are a complete mystery to vast majority of those that are not. It would therefore be highly beneficial to re-structure the manuscript in such a way that the findings can reach the many researchers that could benefit from this powerful approach rather than the few who already use it.

1. Duty cycles, fluorescence saturation, water absorption effects and longer acquisition times lead to greater phototoxicity, so may not be appropriate for most dynamic biological applications where acquisition speed and/or continued image acquisitions are the key factors. The authors should explain and perform cell viability tests to address this in the context of their current analysis approaches. Calcium imaging for assessing tissue viability is not the technique of choice for most readers and is presented in a way that assumes general knowledge that does not exist. Assays using membrane-impermeant DNA dyes, or other live-dead assays are far more common, and this study would benefit significantly from inclusion of these assays. The authors should also clearly explain the need for such viability analysis and clearly discuss the potential limitations of their imaging regimes with respect to live samples.

2. How long does it take to acquire a single frame with four-photon excitation at 1700nm? In none of the data sets was frame time mentioned in particular when acquired 3D data sets. Can the authors please ensure that these times are mentioned both in the main text and the figures containing images.

3. Line131/Fig3D: the authors present data showing relative axial resolution measurements. Are these features measured diffraction limited and how do they know? These are clearly not like for like structures (different fluorescent species) so it is not clear this can be used as a measure of resolution. Can the authors please provide other resolution measurements?

4. Lines 140-42: the authors present data showing the advantages of THG at 1650nm over other excitation lines. Aside from the excitation wavelength could this data be explained by the greater absorption and scattering at the emission wavelengths generated at these laser lines?

5. Figure 3A and C: the SNR for 1650nm increases whilst for 1300nm and 1180 excitation this decreases. Is this simply due to more of the exciting fluorophore species residing deeper into the tissue?

6. The authors should provide a clear view early on in the introduction of the problems associated with imaging biological tissues with the longer wavelengths that are used for confocal laser scanning microscopy and for two-photon laser scanning microscopy and present this in an accessible manner. For most readers, the need for a larger laser won't be their first question; instead, it will be the viability after/during the imaging session. The expected temperature rise, and an indirect mention of burn marks, comes at the end of the section but should be discussed early on so the potential limitations are transparently presented.

7. The findings and the figures in general should be presented in a more accessible and accurate manner. The figures are highly unlikely to be understandable by the readers who do not perform this sort of imaging, and the legends do not help as they are either far too brief or contain too many terms that are poorly explained. The legend and figure that it describes should be able to stand on their own, not simply defer to the body of the paper to explain what was done or is shown. For experts in the field, the figures are also not as convincing as they could be or presented in a way that they can be critically evaluated.

---

## [Author Response]

This is a very interesting study with some potentially important findings from a technical perspective. However, there is a disconnect at present between the quality of the work and the quality of the presentation. There are many areas of quantitative imaging and intravital imaging that are well known to those in the direct field, and that are a complete mystery to vast majority of those that are not. It would therefore be highly beneficial to re-structure the manuscript in such a way that the findings can reach the many researchers that could benefit from this powerful approach rather than the few who already use it.

We rewrote the introduction and discussion for clarity and accessibility. We further discussed the relation between excitation parameters, linear and nonlinear phototoxicity, and cell viability as well as limitations of the method regarding imaging depth and speed in greater detail.

In addition, additional phototoxicity measurements (revised Figure 2) demonstrate the exquisite sensitivity of intracellular calcium measurements to report both reversible and irreversible cell damage, which outperforms endpoint analyses using nuclear dyes.

1. Duty cycles, fluorescence saturation, water absorption effects and longer acquisition times lead to greater phototoxicity, so may not be appropriate for most dynamic biological applications where acquisition speed and/or continued image acquisitions are the key factors. The authors should explain and perform cell viability tests to address this in the context of their current analysis approaches. Calcium imaging for assessing tissue viability is not the technique of choice for most readers and is presented in a way that assumes general knowledge that does not exist. Assays using membrane-impermeant DNA dyes, or other live-dead assays are far more common, and this study would benefit significantly from inclusion of these assays. The authors should also clearly explain the need for such viability analysis and clearly discuss the potential limitations of their imaging regimes with respect to live samples.

We share these concerns regarding phototoxicity associated with excitation at 1700 nm and are aware of the limitations which have been studied for 3-photon excitation in the brain^18^. Accordingly, the discussion has been revised, as follows (p. 12-13):

“However, additional experiments and simulations defining linear phototoxicity limits need to be performed for the 1700 nm excitation range. It remains to be established to what extend the maximum tolerable power depends on differences in water absorption at 1700 nm compared to 1300 nm^10,11^. Tissue heating may be reduced and imaging depth increased by reducing the imaging frame rate, which allows the tissue to cool down between images or lines, or by reducing the frequency of excitation pulses. This is possible when the biological process of interest can be monitored with lower frame rates or image resolution. For example, studying cancer cell invasion only requires image acquisition with 10-min intervals, which allows effective heat dissipation during image acquisition; conversely, high scan rates required for calcium imaging deeply inside the tumor (20 sec) may require a higher repetition rate of excitation pulses and preclude effective deep imaging in highly scattering tissue^8^.”

In our hands, using cells expressing relatively high H2B-eGFP and Hoechst applied at regular dose in vivo, 4-photon excitation was effective at pulse energies <= 9 nJ at the sample surface when focused 50 µm deep (revised Figure 1b and 3a), which is below the phototoxic threshold (revised Figure 2d and 2i). Therefore, we consider 4P excitation as a relevant new microscopic modality in life sciences. As potential shortcoming, we noted that restrictions on labelling density and quantum efficiency will be more severe for 4-photon excited fluorophores as compared to 3-photon excited fluorophores (revised Figure 5b). In summary, 3P and 4P fluorescence excitation will both be suitable for a broad range of live-cell biology applications.

2. How long does it take to acquire a single frame with four-photon excitation at 1700nm? In none of the data sets was frame time mentioned in particular when acquired 3D data sets. Can the authors please ensure that these times are mentioned both in the main text and the figures containing images.

The frame acquisition times have been included in the figure captions. As example, the following can be found in the figure caption of revised Figure 3, panel a: “Details from images: 2.2 s frame time, 20 µs pixel integration time.” This frame rate is sufficient for monitoring cell migration and tissue remodelling, as used in the tumor and immunology fields (addressed above).

3. Line131/Fig3D: the authors present data showing relative axial resolution measurements. Are these features measured diffraction limited and how do they know? These are clearly not like for like structures (different fluorescent species) so it is not clear this can be used as a measure of resolution. Can the authors please provide other resolution measurements?

The quantitative analyses on resolution using diffraction-limited 100 nm beads are presented in revised Figure 1—figure supplement 1d. In revised Figure 3d, the structures analysed for relative resolution measurements (nuclei, cell bodies) were not diffraction limited. Different fluorescent species indeed imply different resolutions, in case the order of the excitation process is different between the species. However, the steep discontinuity gradient of the label at the cell or nucleus edge provides an approximation for detecting the resolution in the axial direction in a discontinuous 3D tissue sample in vivo. We here aimed to provide a simple approach to detect resolution in vivo, given that direct detection of diffraction-limited beads is not possible in scattering fluorescent tumors in mice. We submit, that the steepness in the normalized signal strength over the discontinuity (minimum – maximum between 0 – 1) of different structures can serve as pragmatic measure for the axial resolution.

4. Lines 140-42: the authors present data showing the advantages of THG at 1650nm over other excitation lines. Aside from the excitation wavelength could this data be explained by the greater absorption and scattering at the emission wavelengths generated at these laser lines?

Indeed, THG intensity was strongest at 1650 nm excitation over the other excitation conditions. We did not directly address the mechanism of this difference, using physically controlled conditions.

The improved signal to noise and imaging depth of low repetition rate high pulse energy excitation (1300 nm and 1650 nm) with respect to high repetition rate low pulse energy excitation (1180 nm) has been addressed as follows (p. 11):

“Imaging depth limits of THG microscopy with lowIR excitation results from the lower photon density in the focus, leading to a drastically lower excitation efficiency, in combination with the rapid attenuation of nonlinear signals upon scattering and absorption of excitation light^19,20^. In contrast, THG by highIR excitation, which creates a multifold higher photon density in the focus, is achieved with only a small fraction of the available power at the tissue surface, which allows to increase the excitation energy at the sample surface with imaging depth further, to maintain a constant signal over a much broader depth range.”

The improved THG imaging performance of 1650 nm with respect to 1300 nm high-pulse-energy excitation may be addressed to several causes. Wavelength dependent differences in scattering and absorption of the laser light are the most plausible explanation for this effect. Scattering and absorption are dominant attenuation factors for excitation of THG, which depends on the cube of the excitation energy arriving at the focus, while the emission is much less dependent on these parameters^19,21^. Therefore, shifting the excitation wavelength may strongly modulate the detected signal in dense 3D tissue, since the effective attenuation length of tumor tissue is much longer for 1650 nm excitation as compared to 1300 nm excitation (revised Figure 3e).

In addition, absorption of the emitted light may indeed contribute to the improved THG signal at 1650 nm, because the blood absorption coefficient is 10 times higher at ~400 nm (THG emission at 1200-1300 nm incident wavelength) compared to 550 nm (emission from 1650 nm incident). However, we noted only minor impact of these emission wavelengths on the effective attenuation lengths (*l_e_*), derived for 1300 nm excitation: *l_e_* = 120 µm and *l_e_* = 128 µm were found, for 433 nm (THG) and 500-550 nm (eGFP), respectively (revised Figure 3—figure supplement R1a, shown below). Also, for 1650 nm excitation, we did not find a strong correlation between the emission wavelength and the effective attenuation length: *l_e_* = 218 µm and *l_e_* = 220 µm were found, for 485-525 nm (4-photon excited eGFP) and 570-610 nm (TagRFP), respectively (revised Figure 3—figure supplement R1b). Notably, for 1650 nm we derived a larger effective attenuation length for THG emission centred around 550 nm as compared to fluorescence emission: *l_e_* = 286 µm (revised Figure 3—figure supplement R1b), while this difference was absent for 1300 nm excitation. This finding indicates other excitation wavelength dependent mechanisms are influencing THG excitation efficiency as well, such as the size of THG signal emitting structures relative to the size of the focus.

While technically interesting, we feel that this data goes beyond the primary application focus of this work; however, should the referees and Editor support inclusion in the Supplement, we will be happy to revise accordingly.

In summary, we conclude that the longer excitation wavelength and higher excitation pulse energy are favourable for THG microscopy in the tumor, while the absorption and scattering of the emission are less critical.

Reviewer 1:In this manuscript, the authors show they can accomplish imaging in complex specimens using 3- and 4-photon excitation, deeper in the specimen than comparable optics can accomplish with 2-photon excitation laser scanning microscopy. This is a clear advantage for imaging optically hostile specimens such as cultured organoids or spheroids, or in challenging in vivo settings. I am excited about these findings, but I am not at all supportive of the current version of the manuscript being used to present these lovely findings.There are two strong reasons for my opinion:i. The manuscript presents the findings in a manner that will only be understandable by the readers who are familiar with the topic, and who are likely to already have heard of the capabilities of 3- and 4-photon excitation to image deeper into specimens.ii. The results are not presented in a way that the large body of potential readers can understand. They will be unable to grasp the way that the experiments were performed, or understand what the figures are showing, or critically evaluate the results that are presented.Thus, there is a disconnect between the quality of the work and the quality of the presentation. There are many areas of quantitative imaging and intravital imaging that are well known to those that know about them (or uses them), and that are a complete mystery to vast majority of those that don't know about the tools or use them. The authors must take this as an opportunity to reach the many workers that could benefit from this powerful approach, rather than writing for the group that already knows (and even uses) the approaches presented.

We rewrote large parts of the introduction, added detail to the results to explain each section in greater depth, and expanded the discussion to better address a broad life sciences-oriented audience with affinity for microscopy.

1. Provide needed background and present important things first. The authors should give the reader a clear view into the issues in imaging biological tissues with the longer wavelengths that are used for confocal laser scanning microscopy (CLSM) and for two-photon laser scanning microscopy (TPLSM). There are several factoids presented, all seemingly true, but not presented in an accessible manner. Rather than starting with a mention of the expected temperature rise due to the dramatically higher absorbance by water of 1300nm and 1700nm light, the paper first presents the major absorbance of the light (~2/3 loss) and that this isn't a problem because there is sufficient laser power. For most readers, the need for a larger laser won't be their first question; instead, it will be the viability after/during the imaging session. The expected temperature rise, and an indirect mention of burn marks (!), comes at the end of the section.

The following specific points were revised:

1. The introduction has been rewritten to bring the proposed microscopy method in perspective with state-of-the-art methods used for intravital imaging of biological tissues, with the aim to familiarize a broad, life sciences-oriented audience to this relatively new technology.

2. The Results section has been restructured and new results characterizing phototoxicity with a commonly used viability probe have been integrated with the calcium sensor-based viability measurements. We now explain the principle of 3- and 4-photon excitation in comparison to 2-photon excitation; discuss the advantage of the presented method over the state-of-the-art methods available for intravital imaging of different tissues; application purposes; considerations for safe application, avoiding phototoxicity; and strategies to monitor phototoxicity in tissue models.

3. We added a panel explaining the principles of 1-, 2-, 3- and 4PE and higher harmonics generation (revised Figure 1a).

4. The excitation parameters of confocal LSM, conventional 2-photon microscopy and microscopy with high-pulse-energy low pulse-repetition-rate excitation are now summarized side-by-side (Supplementary File 1a).

5. The phototoxicity paragraph has been extended, to provide sufficient background on the different mechanisms, locations of phototoxicity and their relation with high-pulse-energy excitation. We further discuss approaches to detect phototoxicity, including real-time calcium imaging and commonly used endpoint analysis.

6. THG microscopy performance comparison has been addressed in a separate paragraph. As a consequence, Figures 3 and 4 have been rearranged to include separate fluorescence and THG microscopy-based figures.

7. The maximum imaging depth has been quantified by specifying a detection limit for the signal to noise ratio, to better support the sections on imaging depth improvements of 3-photon microscopy over conventional 2-photon microscopy.

8. New Figure 5 discusses 3- and 4-photon excitation modalities in comparison to conventional 2-photon microscopy.

9. Technical details were moved from the text body to supplementary images and/or to the *Materials and methods* section, including detailed setup resolutions and pixel dwell times.

2. Explain and perform cell viability tests. Calcium imaging for assessing tissue viability is not the technique of choice for most readers, and is presented in a way that assumes general knowledge that simply does not exist. Membrane patency assays using membrane-impermeant DNA dyes, or other live-dead assays are far more common, but not presented in this study. I am not insistent that the authors use any particular assay, but I am insistent that the authors present the need for viability assay(s), teach the reader the principles of the assay(s) used, and present the results in an understandable manner.

We explained the need for viability analyses more clearly and extended the background on phototoxicity as well as the rationale for calcium sensor for in vivo monitoring. We performed additional live cell experiments, showing the superior sensitivity and responsiveness of the calcium sensor with respect to membrane patency assays (Sytox green). Both the calcium sensor and Sytox green showed phototoxicity effects for excitation pulse energies above the same threshold (new Figure 2—figure supplement 1e), however calcium but not Sytox detection allows to quantify reversible phototoxic stress. Furthermore, the threshold for nonlinear phototoxicity (phototoxicity limited to the focal plane) was updated, including statistical analysis (p. 8 and revised Figure 2d).

We discuss the required pulse energies and safety thresholds as follows (p. 13): “Cell stress and bleaching were induced in the focal plane when pulse energies reached threshold values (3.9 nJ for 1300 nm and 12 nJ for 1650 nm at 50 µm depth), eventually leading to physical disruption of tissue structures (burning marks), in line with observations by others^1^. Conversely, energy doses below the thresholds were tolerated by cells without in-focus ca^2+^ elevation, and this energy level was sufficient to achieve high-quality multi-parametric image acquisition at matched depth. The in-focus phototoxicity thresholds obtained in the cell-rich tumor model are comparable to the dynamic range detected in low-density tissues^2–4^, indicating that nonlinear phototoxicity might have a general cause, such as light-induced ionization or formation of reactive oxygen species,^5,6^ irrespective of the tissue density.”

Furthermore, we discussed what experimental conditions may require to be further explored, to control for subtle effects on cell biology (p. 13): “The impact of sub-threshold highIR light exposure on long-term integrity of cell structure and function requires further exploration, including growth, differentiation and, chromatin integrity^7^, as well as benchmarking of stress-sensitive complex cell functions, such as cell migration and cell-cell interactions^8,9^.”

Lastly, we discuss how phototoxicity and signal to noise ratios within the acquired images restrict imaging speed and depth, as follows (p. 12-13): “However, additional experiments and simulations defining linear phototoxicity limits need to be performed for the 1700 nm excitation range. It remains to be established to what extend the maximum tolerable power depends on differences in water absorption at 1700 nm compared to 1300 nm^10,11^. Tissue heating may be reduced and imaging depth increased by reducing the imaging frame rate, which allows the tissue to cool down between images or lines, or by reducing the frequency of excitation pulses. This is possible when the biological process of interest can be monitored with lower frame rates or image resolution. For example, studying cancer cell invasion only requires image acquisition with 10-min intervals, which allows effective heat dissipation during image acquisition; conversely, high scan rates required for calcium imaging deeply inside the tumor (20 sec) may require a higher repetition rate of excitation pulses and preclude effective deep imaging in highly scattering tissue^8^.”

3. Present the finding and the figures in an accessible manner. The figures are simply not digestible by the readers who do not perform this sort of work, and the legends do not help sufficiently. For those of us who do perform work of this sort, the figures are not as convincing as they should be, or presented in a way that they can be critically evaluated.

Figures and captions have been revised to address a broad readership. Short titles explaining the main message of each subpanel have been included. As example, the revised cation of Figure 1 states: “Multimodal microscopy of fluorescent skin tumor xenografts in vivo showing up to 4 fluorescent and 2 label-free tissue morphology channels, simultaneously excited with a single highIR wavelength. (a) Jablonski diagrams (left) and scaled excitation / emission spectra (right) of 1-, 2-, 3- and 4P-excited fluorescence and higher harmonic generation. … (b) 5-channel 1300 nm excited (left) and 6-channel 1650 nm excited (middle and right) images were taken in the center of fluorescent tumors through a dermis imaging window. Images were selected from median-filtered (1 pixel) z-stacks. The excitation wavelength, calculated pulse energy at the sample surface and imaging depth z are shown (top). Cell nuclei containing a mixture of mCherry and Hoechst appear as green (right). Details from images left to right: … (c) Zoomed xy-plane with individual channels (left) and orthogonal (xz-) projection (right) from (b), dotted rectangle. (d) Relation between measured emissions and excitation energy reveals the order of the excitation processes. …”.

Consider the legend for Figure 1:"Microscopy with simultaneous 2-, 3- and 4 photon processes excited in fluorescent skin tumor xenografts in vivo. Representative images were selected from median-filtered (1 pixel) z-stacks, which were taken in the center of fluorescent tumors through a dermis imaging window. a) Excitation at 1300nm (OPA) in day-10 tumor at 145 μm imaging depth with a calculated 3.3 nJ pulse energy at the sample surface, 24 μs pixel integration time and 0.36 μm pixel size. For calculation of pulse energy at the sample surface see Figure S3. b) Excitation at 1650 nm (OPA) in day-13 tumor at 30 μm depth with a calculated 6.3 nJ pulse energy at the sample surface, 12 μs pixel integration time and 0.46 μm pixel size. c) Excitation at 1650 nm (OPA) in day-14 tumor at 85 μm depth, with a calculated 5.4 nJ pulse energy at the sample surface, 12 μs pixel integration time and 0.46 μm pixel size. Cell nuclei containing a mixture of mCherry and Hoechst appear as green."If I gave any of the figures and legends to the people in my lab, the half that don't do multi-photon imaging (but that have sat through many lab meetings) would just hand them back to me with quizzical expressions on their faces.The figures are not as compelling as the results, and defer to the body of the paper to explain what was done or what was shown, and assumes that the average reader remembers the differences between OPO and OPA , for example (which they won't). The power plots showing nJ and mW in Figure 3 are inaccessible to most reader, and not well described.

See also previous reply. Abbreviations and acronyms introduced in the text body are now integrated in the figures. The power plots have been revised to be self-explanatory and the relation between excitation power and imaging depth has been updated, as follows (p. 36, Figure 3 figure caption): “Z-stacks were recorded with increasing excitation pulse energy E at the sample surface with increasing imaging depth z (grey profiles right from orthogonal views).”

I should mention that the figures, legends and text are not satisfying for the readers who are familiar with 2-, 3- and 4-photon imaging either. These are fantastic findings, and deserve figures that are as lovely as the results, and are compelling. Some of these issues are due to typos:"Consistently, multiparameter recordings were achieved inside the tumor at 350 μm depth using excitation at 1650 nm and 1300 nm, but 1180 nm (Figure 3b). "

Typos have been corrected. The example given above has been corrected, as follows (p. 9): “Multiparameter recordings were reliably obtained at a depth of 350 µm inside the tumor using 1300 and 1650 nm highIR excitation, but not with the 1180 nm lowIR source (Figure 3b).”.

The scaling of part of the panels has been adapted to render the figures more balanced.

However, the greater problem is that the text doesn't present the findings in a straightforward, convincing fashion and then interpret them. Instead, the conclusion often leads the evidence:"In line with an improved depth range, the signal-to-noise ratio (SNR) of 3PE TagRFP outperformed the SNR of 2PE TagRFP at depths beyond 150 μm (Figure 3c). Because H2B-eGFP expression in HT1080 tumors was very high, 3PE eGFP emission reached the highest SNR."

We agree and have restructured each Results section with the sequence background / aim, qualitative and quantitative results, concluding statement.

The legend and figure that it describes should be able to stand on their own, and convince a skeptical reader with the help of the text in the body of the manuscript.

We have revised the images and captions accordingly. For example, the addition of Figure panel 1a now provides additional background to understand the nonlinear excitation processes behind panel 1b, and explains the abbreviations 1P-4P, SHG and THG which are used in all images.

Reviewer 3:1. It seems as though when you take into consideration duty cycles, fluorescence saturation, water absorption effects and longer acquisition times, which lead to greater phototoxicity, 4-PE at 1700nm excitation is not appropriate for most dynamic biological applications where acquisition speed and/or continued image acquisitions are the key factors. Could the authors comment on this?

We share these concerns regarding phototoxicity associated with excitation at 1700 nm and are aware of the limitations which have been studied for 3-photon excitation in the brain^18^. Accordingly, the discussion has been revised, as follows (p. 12-13): “However, additional experiments and simulations defining linear phototoxicity limits need to be performed for the 1700 nm excitation range. It remains to be established to what extend the maximum tolerable power depends on differences in water absorption at 1700 nm compared to 1300 nm^10,11^. Tissue heating may be reduced and imaging depth increased by reducing the imaging frame rate, which allows the tissue to cool down between images or lines, or by reducing the frequency of excitation pulses. This is possible when the biological process of interest can be monitored with lower frame rates or image resolution. For example, studying cancer cell invasion only requires image acquisition with 10-min intervals, which allows effective heat dissipation during image acquisition; conversely, high scan rates required for calcium imaging deeply inside the tumor (20 sec) may require a higher repetition rate of excitation pulses and preclude effective deep imaging in highly scattering tissue^8^.”

In our hands, using cells expressing relatively high H2B-eGFP and Hoechst applied at regular dose in vivo, 4-photon excitation was effective at pulse energies <= 9 nJ at the sample surface when focused 50 µm deep (revised Figure 1b and 3a), which is below the phototoxic threshold (revised Figure 2d and 2i). Therefore, we consider 4P excitation as a relevant new microscopic modality in life sciences. As potential shortcoming, we noted that restrictions on labelling density and quantum efficiency will be more severe for 4-photon excited fluorophores as compared to 3-photon excited fluorophores (revised Figure 5b). In summary, 3P and 4P fluorescence excitation will both be suitable for a broad range of live-cell biology applications.

2. How long does it take to acquire a single frame with four-photon excitation at 1700nm? In none of the data sets was frame time mentioned in particular when acquired 3D data sets. Can the authors ensure that these times are mentioned both in the main text and the figures containing images.

The frame acquisition times have been included in the figure captions. As example, the following can be found in the figure caption of revised Figure 3, panel a: “Details from images: 2.2 s frame time, 20 µs pixel integration time.” This frame rate is sufficient for monitoring cell migration and tissue remodelling, as used in the tumor and immunology fields (addressed above).

3. In line 131 and figure 3d the authors present data showing relative axial resolution measurements. Are these features measured diffraction limited and how do they know? There are clearly not measuring like for like structures (different fluorescent species) so do not think this can be used as a measure of resolution. Can the author provide other resolution measurements?

The quantitative analyses on resolution using diffraction-limited 100 nm beads are presented in revised Figure 1—figure supplement 1d. In revised Figure 3d, the structures analysed for relative resolution measurements (nuclei, cell bodies) were not diffraction limited. Different fluorescent species indeed imply different resolutions, in case the order of the excitation process is different between the species. However, the steep discontinuity gradient of the label at the cell or nucleus edge provides an approximation for detecting the resolution in the axial direction in a discontinuous 3D tissue sample in vivo. We here aimed to provide a simple approach to detect resolution in vivo, given that direct detection of diffraction-limited beads is not possible in scattering fluorescent tumors in mice. We submit, that the steepness in the normalized signal strength over the discontinuity (minimum – maximum between 0 – 1) of different structures can serve as pragmatic measure for the axial resolution.

4. In line 140 – 142 the authors present data showing the advantages of THG at 1650nm over other excitation lines. Aside from the excitation wavelength could this data be explained by the greater absorption and scattering at the emission wavelengths generated at these laser lines?

Indeed, THG intensity was strongest at 1650 nm excitation over the other excitation conditions. We did not directly address the mechanism of this difference, using physically controlled conditions.

The improved signal to noise and imaging depth of low repetition rate high pulse energy excitation (1300 nm and 1650 nm) with respect to high repetition rate low pulse energy excitation (1180 nm) has been addressed as follows (p. 11): “Imaging depth limits of THG microscopy with lowIR excitation results from the lower photon density in the focus, leading to a drastically lower excitation efficiency, in combination with the rapid attenuation of nonlinear signals upon scattering and absorption of excitation light^19,20^. In contrast, THG by highIR excitation, which creates a multifold higher photon density in the focus, is achieved with only a small fraction of the available power at the tissue surface, which allows to increase the excitation energy at the sample surface with imaging depth further, to maintain a constant signal over a much broader depth range.”

The improved THG imaging performance of 1650 nm with respect to 1300 nm high-pulse-energy excitation may be addressed to several causes. Wavelength dependent differences in scattering and absorption of the laser light are the most plausible explanation for this effect. Scattering and absorption are dominant attenuation factors for excitation of THG, which depends on the cube of the excitation energy arriving at the focus, while the emission is much less dependent on these parameters^19,21^. Therefore, shifting the excitation wavelength may strongly modulate the detected signal in dense 3D tissue, since the effective attenuation length of tumor tissue is much longer for 1650 nm excitation as compared to 1300 nm excitation (revised Figure 3e).

In addition, absorption of the emitted light may indeed contribute to the improved THG signal at 1650 nm, because the blood absorption coefficient is 10 times higher at ~400 nm (THG emission at 1200-1300 nm incident wavelength) compared to 550 nm (emission from 1650 nm incident). However, we noted only minor impact of these emission wavelengths on the effective attenuation lengths (*l_e_*), derived for 1300 nm excitation: *l_e_* = 120 µm and *l_e_* = 128 µm were found, for 433 nm (THG) and 500-550 nm (eGFP), respectively (revised Figure 3—figure supplement 1a). Also, for 1650 nm excitation, we did not find a strong correlation between the emission wavelength and the effective attenuation length: *l_e_* = 218 µm and *l_e_* = 220 µm were found, for 485-525 nm (4-photon excited eGFP) and 570-610 nm (TagRFP), respectively (revised Figure 3—figure supplement 1b). Notably, for 1650 nm we derived a larger effective attenuation length for THG emission centred around 550 nm as compared to fluorescence emission: *l_e_* = 286 µm (revised Figure 3—figure supplement 1b), while this difference was absent for 1300 nm excitation. This finding indicates other excitation wavelength dependent mechanisms are influencing THG excitation efficiency as well, such as the size of THG signal emitting structures relative to the size of the focus.

While technically interesting, we feel that this data goes beyond the primary application focus of this work; however, should the referees and Editor support inclusion in the Supplement, we will be happy to revise accordingly.

In summary, we conclude that the longer excitation wavelength and higher excitation pulse energy are favourable for THG microscopy in the tumor, while the absorption and scattering of the emission are less critical.

5. In figure 3A and 3C the SNR for 1650nm increases whilst for 1300nm and 1180 excitation this decreases. Is this simply due to more of the exciting fluorophore species residing deeper into the tissue?

Indeed, the slope of the curves shown in Figure 3C is bimodal. We escalated the excitation power gradually to achieve constant emission with increasing tissue penetration, and this correction was done “by eye” through the operator. Likely, this introduced some overcompensation and, accordingly, inaccuracy. We now describe the procedure, as follows (p. 9): “To achieve arbitrarily constant 3- or 2P emission with increasing imaging depth, we manually increased the excitation energy E using eGFP (for 1300 nm) and TagRFP (1180 and 1650 nm) as a reference, while remaining below the phototoxicity thresholds as defined above (Figure 3a, grey profiles).”

In Figure 3G (Figure 4B in the revised manuscript), different structures with different dimensions and orientations responded differently to the three excitation modalities. THG from thin fibrous collagen structures near the tumor surface was less efficiently excited with 1650 nm compared to shorter excitation wavelengths. On the other hand, cell and tissue structures within the tumor mass were excited more effectively with 1650 nm as compared to the collagen structures. At imaging depths beyond ~300 µm, excitation energy at the surface was not further increased, causing a decrease of the THG signal. As a consequence, the 1650 nm THG signal curve shows a peak. The thin fibrous collagen structures responded more strongly to 1180 nm and 1300 nm excitation, while cell and tissue structures inside the tumor emitted a relatively lower signal at these wavelengths. Therefore, 1180 and 1300 nm excited THG curves show a slope.

We now explain these differences in THG responses in greater detail, as follows (p. 11): “For example, THG by small collagen fibers under non-scattering conditions was best with excitation at 1180 nm, whereas THG by tumor cell nuclei was strongest at 1300 nm excitation (Figure 4a, arrowheads). The wavelength-dependent size of the focal volume relative to THG-generating cell and tissue structures is a possible explanation for these differences^17^.”

References:

1. Hopt, a and Neher, E. Highly nonlinear photodamage in two-photon fluorescence microscopy. *Biophys. J.* 80, 2029–36 (2001).

2. Ouzounov, D. G. *et al.* In vivo three-photon imaging of activity of GCaMP6-labeled neurons deep in intact mouse brain. *Nat. Methods* 14, 388–390 (2017).

3. Yildirim, M., Sugihara, H., So, P. T. C. and Sur, M. Functional imaging of visual cortical layers and subplate in awake mice with optimized three-photon microscopy. *Nat. Commun.* 10, 177 (2019).

4. Wang, T. *et al.* Quantitative analysis of 1300-nm three-photon calcium imaging in the mouse brain. *eLife* 9, 1–22 (2020).

5. Débarre, D. *et al.* Mitigating phototoxicity during multiphoton microscopy of live drosophila embryos in the 1.0-1.2 μm wavelength range. *PLoS One* 9, (2014).

6. Tirlapur, U. K., König, K., Peuckert, C., Krieg, R. and Halbhuber, K. Femtosecond near-infrared laser pulses elicit generation of reactive oxygen species in mammalian cells leading to apoptosis-like death. *Exp. Cell Res.* 263, 88–97 (2001).

7. Débarre, D., Olivier, N., Supatto, W. and Beaurepaire, E. Mitigating phototoxicity during multiphoton microscopy of live drosophila embryos in the 1.0-1.2 μm wavelength range. *PLoS One* 9, (2014).

8. Weigelin, B. *et al.* Cytotoxic T cells are able to efficiently eliminate cancer cells by additive cytotoxicity. *Nat. Commun.* 12, 1–12 (2021).

9. Kiepas, A., Voorand, E., Mubaid, F., Siegel, P. M. and Brown, C. M. Optimizing live-cell fluorescence imaging conditions to minimize phototoxicity. *J. Cell Sci.* 133, (2020).

10. Cheng, L.-C., Horton, N. G., Wang, K., Chen, S.-J. and Xu, C. Measurements of multiphoton action cross sections for multiphoton microscopy. *Biomed. Opt. Express* 5, 3427–33 (2014).

11. Horton, N. G. *et al.* In vivo three-photon microscopy of subcortical structures within an intact mouse brain. *Nat. Photonics* 7, 205–209 (2013).

12. Drobizhev, M., Makarov, N. S., Tillo, S. E., Hughes, T. E. and Rebane, A. Two-photon absorption properties of fluorescent proteins. *Nat. Methods* 8, 393–9 (2011).

13. Hontani, Y., Xia, F. and Xu, C. Multicolor three-photon fluorescence imaging with single-wavelength excitation deep in mouse brain. *Sci. Adv.* 7, (2021).

14. Podgorski, K. and Ranganathan, G. Brain heating induced by near-infrared lasers during multiphoton microscopy. *J. Neurophysiol.* 116, 1012–1023 (2016).

15. Wang, T. *et al.* Three-photon imaging of mouse brain structure and function through the intact skull. *Nat. Methods* 15, 789–792 (2018).

16. Rowlands, C. J. *et al.* Wide-field three-photon excitation in biological samples. *Light Sci. Appl.* 6, e16255-9 (2017).

17. Debarre, D. *et al.* Imaging lipid bodies in cells and tissues using third-harmonic generation microscopy. *Nat. Methods* 3, 47–53 (2006).

18. Li, B., Wu, C., Wang, M., Charan, K. and Xu, C. An adaptive excitation source for high-speed multiphoton microscopy. *Nat. Methods* 17, 163–166 (2020).

19. Beaurepaire, E., Oheim, M. and Mertz, J. Ultra-deep two-photon fluorescence excitation in turbid media. *Opt. Commun.* 188, 25–29 (2001).

20. Theer, P., Hasan, M. T. and Denk, W. Two-photon imaging to a depth of 1000 µm in living brains by use of a Ti:Al_2O_3 regenerative amplifier. *Opt. Lett.* 28, 1022 (2003).

21. Theer, P. and Denk, W. On the fundamental imaging-depth limit in two-photon microscopy. *J. Opt. Soc. Am. A. Opt. Image Sci. Vis.* 23, 3139–49 (2006).

22. Wang, M. *et al.* Comparing the effective attenuation lengths for long wavelength in vivo imaging of the mouse brain. *Biomed. Opt. Express* 9, 3534 (2018).